# Evolutionary insights into primate skeletal gene regulation using a comparative cell culture model

**Genevieve Housman** [1]*, **Emilie Briscoe** [1], **Yoav Gilad** [1,2]

**1** Section of Genetic Medicine, Department of Medicine, University of Chicago, Chicago, Illinois, United States of America, **2** Department of Human Genetics, University of Chicago, Chicago, Illinois, United States of America

* ghousman@uchicago.edu

## Abstract

The evolution of complex skeletal traits in primates was likely influenced by both genetic and environmental factors. Because skeletal tissues are notoriously challenging to study using functional genomic approaches, they remain poorly characterized even in humans, let alone across multiple species. The challenges involved in obtaining functional genomic data from the skeleton, combined with the difficulty of obtaining such tissues from nonhuman apes, motivated us to consider an alternative *in vitro* system with which to comparatively study gene regulation in skeletal cell types. Specifically, we differentiated six human (*Homo sapiens*) and six chimpanzee (*Pan troglodytes*) induced pluripotent stem cell lines (**iPSCs**) into mesenchymal stem cells (**MSCs**) and subsequently into osteogenic cells (bone cells). We validated differentiation using standard methods and collected single-cell RNA sequencing data from over 100,000 cells across multiple samples and replicates at each stage of differentiation. While most genes that we examined display conserved patterns of expression across species, hundreds of genes are differentially expressed (**DE**) between humans and chimpanzees within and across stages of osteogenic differentiation. Some of these interspecific DE genes show functional enrichments relevant in skeletal tissue trait development. Moreover, topic modeling indicates that interspecific gene programs become more pronounced as cells mature. Overall, we propose that this *in vitro* model can be used to identify interspecific regulatory differences that may have contributed to skeletal trait differences between species.

## Author summary

Primates display a range of skeletal morphologies and susceptibilities to skeletal diseases, but the molecular basis of these phenotypic differences is unclear. Studies of gene expression variation in primate skeletal tissues are extremely restricted due to the ethical and practical challenges associated with collecting samples. Nevertheless, the ability to study gene regulation in primate skeletal tissues is crucial for understanding how the primate skeleton has evolved. We therefore developed a comparative primate skeletal cell culture

**Data Availability Statement:** All scRNA-seq data have been deposited in the NCBI's Gene Expression Omnibus under the SuperSeries accession number GSE181744, which contains the SubSeries GSE167240 and GSE18174. All

computational scripts and analysis pipelines can be found on GitHub at https://github.com/ghousman/human-chimp-skeletal-scRNA.

**Funding:** This project received funding from NIH | National Institute of General Medical Sciences (NIGMS), grant number: R01GM122930 to Y.G.; and from NIH | National Institute of Arthritis and Musculoskeletal and Skin Diseases (NIAMS), grant number: F32AR075397 to G.H. The funders had no role in study design, data collection and analysis, decision to publish, or preparation of the manuscript.

**Competing interests:** The authors have declared that no competing interests exist.

model that allows us to access a spectrum of human and chimpanzee cell types as they differentiate from stem cells into bone cells. While most gene expression patterns are conserved across species, we also identified hundreds of differentially expressed genes between humans and chimpanzees within and across stages of differentiation. We also classified cells by osteogenic stage and identified additional interspecific differentially expressed genes which may contribute to skeletal trait differences. We anticipate that this model will be extremely useful for exploring questions related to gene regulation variation in primate bone biology and development.

## Introduction

The skeleton is a biologically and evolutionarily important organ system that consists of several tissues, including bone and cartilage. The skeletal system serves a variety of functions, most notably supporting body weight and facilitating locomotion. While these broad functions are conserved across vertebrates, different species have developed distinct skeletal morphologies, which enable differential use of skeletal elements. For instance, certain bony feature shapes and sizes enable efficient bipedal locomotion in humans, while others enable efficient quadrupedal locomotion in certain nonhuman primates [1,2]. Primates also vary in their susceptibilities to different skeletal disorders, such as osteoarthritis [3–7] and osteoporosis [8–10].

The emergence of conserved and divergent skeletal phenotypes within the primate lineage is not fully resolved. Clarifying the mechanisms that contribute to such complex traits will improve our understanding of skeletal evolution and development. As with all complex traits, skeletal traits are affected by both genetic [11–15] and environmental factors [16–20], and these effects may be mediated, at least in part, through gene expression changes. While the contribution of environmental factors to skeletal differences has been widely studied in the fields of comparative anatomy, forensics, and paleoanthropology, molecular variation in skeletal tissues is not well characterized, especially among primates.

Studying gene expression in skeletal tissues is challenging, as accessing bone and cartilage requires invasive procedures. In addition to the ethical and experimental challenges of collecting skeletal samples from living primates, the poor storage conditions of most skeletal remains that are available, make such samples unusable for most functional genomic applications, including the collection of gene expression data. Further, when well-preserved samples are available, the high cellular heterogeneity of tissues limits data interpretations. Perhaps because of these considerations, even large human transcriptomics consortia like GTEx [21] do not include data from bone and cartilage. Indeed, studies of human skeletal transcriptomics are limited to more targeted efforts to understand skeletal disease, and focus primarily on cartilage tissues and chondrogenic cell types [22–24]. Hence, while comparative primate functional genomics is a growing area of research [25], only a few groups have examined gene regulation in primate skeletal tissues [26–29]. Due to preservation issues, these studies predominantly focus on DNA methylation patterns as opposed to gene expression patterns.

As an alternative to *in vivo* skeletal tissues, induced pluripotent stem cell (**iPSC**) derived cell culture systems provide a new way to explore molecular variation in skeletal cell types. Previous studies have differentiated primate iPSCs into cranial neural crest cells (**CNCCs**), which are precursor cells that develop into a variety of tissues in the skull [29,30]. Additionally, protocols that differentiate iPSCs into osteoblasts [31–33], which are the primary cells in bone, do exist. However, most studies utilizing skeletal cell differentiation schemes include only 1–2 cell lines from humans or other model organisms, and often do not account for the purity of

primary or differentiated cell cultures. Such study designs limit the evolutionary perspective and interpretation of the data generated.

To examine skeletal gene expression among primates more effectively, we have established a comparative primate skeletal cell culture model that includes a large number of human (*Homo sapiens*) and chimpanzee (*Pan troglodytes*) iPSCs. Using this system, we collected and characterized single-cell RNA sequencing (**scRNA-seq**) data from different stages during differentiation towards osteogenic cells. Our study design allowed us to identify interspecific differences in gene expression, which may contribute to skeletal trait divergence between species.

## Results

### Comparative primate study design, data collection, and preprocessing

To study primate skeletal gene expression patterns, we differentiated previously characterized and validated iPSCs from six humans and six chimpanzees [34–39] through an intermediary mesenchymal stem cell (**MSC**) state, and subsequently into osteogenic cells, which are the primary cells in bone (**Figs 1, S1, and S2, Methods, S1–S8 Tables, and S1 Text**). Using the 10X Genomics platform, we measured single-cell gene expression patterns from each cell line at each major stage along this differentiation trajectory–in pluripotent cells (Time 0), mesenchymal cells (Time 1), and osteogenic cells (Time 2) (**Fig 1A, Methods, S9 Table, and S1 Text**). To examine technical reproducibility, we also collected single-cell gene expression data from one human replicate and one chimpanzee replicate at the same stages of differentiation (**Fig 1C**). For each 10X collection, one human cell line and one chimpanzee cell line from the same stage of differentiation were pooled together to ensure that species and batch were not confounded (**S3 Fig**).

Our scRNA-seq approach targeted equal numbers of cells from each species, individual, and replicate. After data processing and filtering (**Methods and S1 Text**), we retained high-quality data from 101,000 cells (**Fig 2A**), with an average of 7,214 cells per individual replicate, a median of 16,157 UMI counts per cell, and a median of 3,929 genes per cell. We assigned data from individual cells to their species of origin using Cell Ranger [40] with certain modifications to accommodate for the low genetic divergence between humans and chimpanzees (see **Methods, S4 and S5 Figs, and S1 Text**) and confirmed that scRNA-seq data are balanced across species (**Fig 2B, 2C, and 2D**). A Uniform Manifold Approximation Projection (**UMAP**) [41] plot indicates that species are fairly well integrated within each stage of differentiation (**S6A Fig**), as expected. Cell counts, UMI counts per cell, and genes per cell for osteogenic cells are slightly lower than those for pluripotent cells and mesenchymal cells, which is likely due to the increased adhesion properties of osteogenic cells. Indeed, osteogenic cells required more potent dissociation reagents (**Methods and S1 Text**) which resulted in lower cell viability (**S3B, S3C, and S7 Figs**) and an increased production of multiplets during 10X runs (**S3E Fig**).

Using data from the replicate samples described above, we found that scRNA-seq data recovery from our cell differentiations is highly reproducible. Specifically, gene expression patterns recovered from different technical replicates are strongly correlated within each stage of differentiation (**Fig 2E, 2F, and 2G, Methods, and S1 Text**), and we observed similar trends when examining data from the two species independently (**S6B Fig**). Lastly, for downstream analyses, we integrated data from all cells using reciprocal principal components analysis (**PCA**) in Seurat [42,43] (**Methods and S1 Text**).

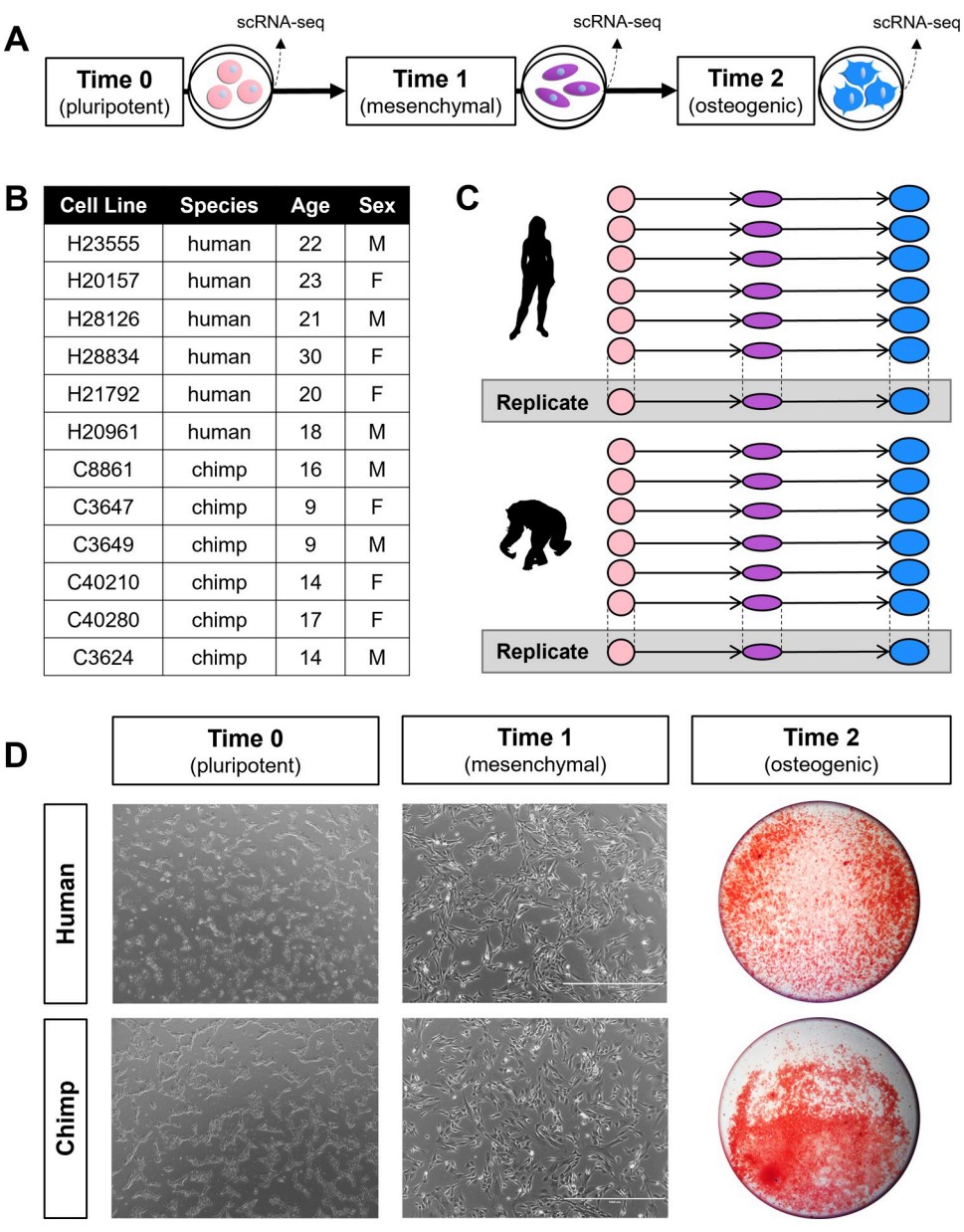

**Fig 1. Comparative skeletal cell culture model.** Schematic of the differentiation protocol and the stages at which single-cell RNA-seq data were collected (A), along with descriptions of the human and chimpanzee cell lines used (B), a diagram of the overall study design (C), and cell images from one human cell line and one chimpanzee cell line at each stage of differentiation (D). Pluripotent cells (Time 0) and mesenchymal cells (Time 1) are phase contrast images at 4X magnification, and osteogenic cells (Time 2) are stained with Alizarin Red and zoomed out to display the entire cell culture well. Silhouette images were adapted from http://phylopic.org/ and courtesy of T. Michael Keesey and Tony Hisgett (http://creativecommons.org/licenses/by/3.0/).

## iPSC-based system effectively models primate osteogenesis

Using standard methods, we validated that iPSCs successfully differentiated into MSCs and osteogenic cells. In accordance with the International Society for Cellular Therapy's minimum criteria for MSCs [44], we visually confirmed that MSCs developed elongated morphologies and were plastic-adherent (**Figs 1D and S1A**). We also used flow cytometry to ensure that

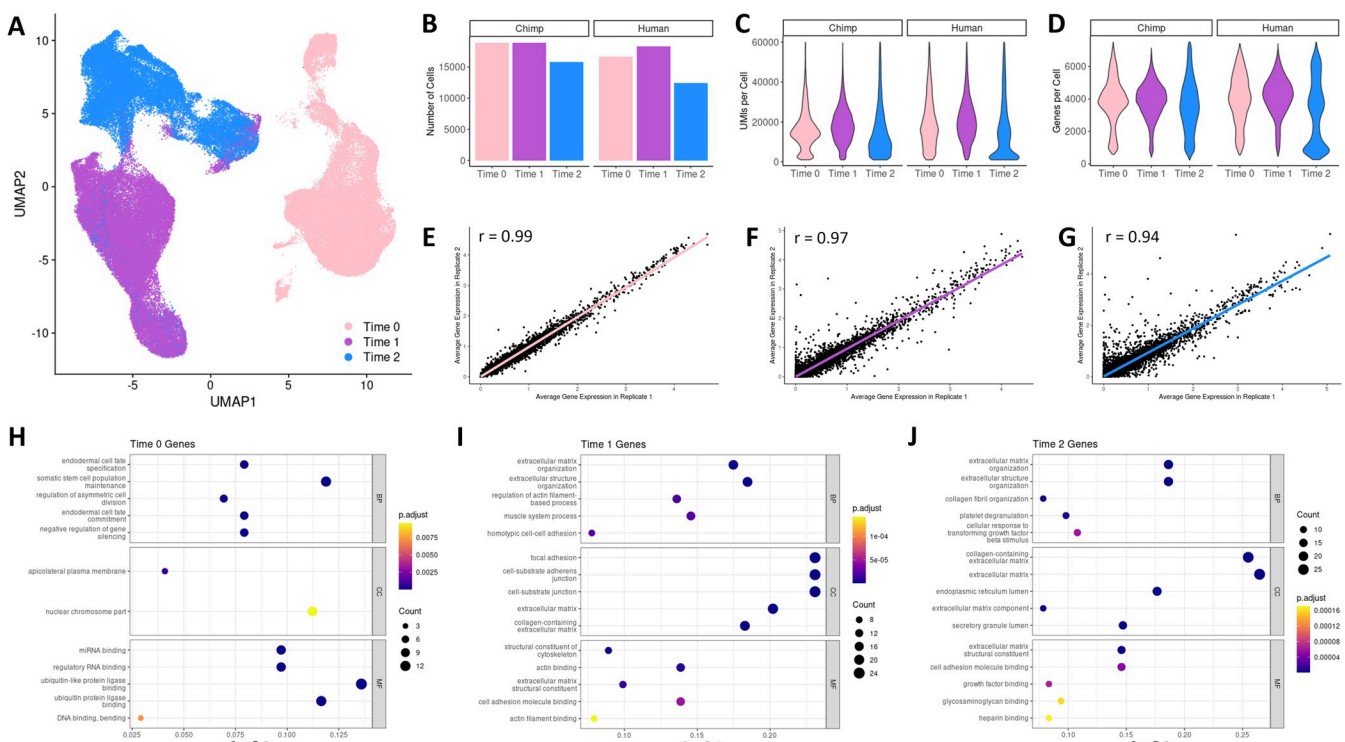

**Fig 2. scRNA-seq data recovery is similar across species and reproducible across replicates.** UMAP dimensional reduction plot of scRNA-seq data with cells labeled by the stage of differentiation at which they were collected (A), along with a bar plot depicting the number of chimpanzee and human cells collected at each stage of differentiation (B), violin plots displaying the distribution of UMI counts per cell (C) and gene counts per cell (D), and the correlation of average gene expression patterns between technical replicates (human and chimpanzee) collected in pluripotent cells (Time 0) (E), mesenchymal cells (Time 1) (F), and osteogenic cells (Time 2) (G). Enrichment of GO functional categories among marker genes for cells collected at each stage of differentiation (H-J). The top 5 GO functions identified in biological processes (BP), cell components (CC), and molecular functions (MF) are displayed along with the adjusted p-value (p-adjust), the number of marker genes overlapping a GO function (Count), and the ratio of overlapping to non-overlapping marker genes for a given GO function (GeneRatio).

MSCs began expressing known cell surface markers (**S1B and S1C Fig** and **S1 Text**). Further, we verified the multipotent differentiation potential of MSCs by performing osteogenic, adipogenic, and chondrogenic differentiations in each cell line (**S1D and S1E Fig** and **S1 Text**). Finally, for osteogenic differentiations, we validated that cells produced extracellular matrices containing calcium deposits, similar to those in bone tissues, using standard Alizarin Red staining (**Figs 1D** and **S2** and **S1 Text**).

Using our processed and integrated scRNA-seq data, we confirmed that cells collected at different stages of differentiation display expected gene expression patterns for known pluripotent, mesenchymal, and osteogenic marker genes. In both species we found that the pluripotency marker *POU5F1* is highly expressed in pluripotent cells, the mesenchymal marker *CD44* is highly expressed in mesenchymal cells and tapers off in osteogenic cells, and the osteogenic marker *COL1A1* is expressed in mesenchymal cells and increases its expression in osteogenic cells (**S6C Fig**).

More broadly, we ensured that cells collected at different stages of differentiation display distinct transcriptomes. First, cell groupings in a UMAP dimensional reduction plot of our scRNA-seq data show that cells collected at different stages of differentiation generally form distinct groups, with some spillover between mesenchymal cells and osteogenic cells (**Fig 2A**). Second, we observed similar transcriptome deviations when performing pairwise correlations

between the whole transcriptome pseudobulk of each stage of differentiation (**S8 Fig** **and** **S1 Text**). Finally, we identified positively expressed marker genes for cells at each stage of differentiation (**Methods**). Testing for the enrichment of gene ontology (**GO**) functional categories in these marker gene sets (**Methods**) revealed expected and biologically relevant functional enrichments, including somatic stem cell population maintenance in pluripotent cells, focal adhesion and extracellular matrix organization in mesenchymal cells, and collagen fibril organization and extracellular matrix production in osteogenic cells (**Fig 2H, 2I, and 2J**).

We also annotated cells using unsupervised clustering and ad hoc assignment methods (**Methods and S1 Text**). These alternative classification schemes produced similar groupings of pluripotent, mesenchymal, and osteogenic cells that displayed similar transcriptomic changes across cell types (**S8**–**S13 Figs**). Thus, we continued using pluripotent, mesenchymal, and osteogenic cell classifications in downstream analyses.

### Identification of differentially expressed genes in our system

Having established our iPSC-derived cell culture system as a reasonable model for studying primate osteogenesis, we then sought to understand how and to what extent the process of osteogenic differentiation differs between humans and chimpanzees. To do this, we first generated pseudobulk data by consolidating single-cell data originating from the same individual, replicate, and cell classification (**Methods**). Using the framework of a linear mixed model to account for the effects of species and cell line (**Methods and Eq 1**), we analyzed pseudobulk data to identify differentially expressed genes between humans and chimpanzees (interspecific DE genes) across each stage of differentiation. Initially, we defined differentiation stage relatively broadly, labeling cells as either pluripotent, mesenchymal, or osteogenic, as described above. Later, we took advantage of our single-cell data to study osteogenesis at a higher resolution, using two alternative approaches–an ad hoc candidate gene-based cell classification approach and a topic modeling strategy.

We discovered hundreds of interspecific DE genes within each stage of differentiation (**Fig 3**). Using standard DE analyses of pseudobulk data (**Methods and S10 Table**), we found 2,098 interspecific DE genes in pluripotent cells, 904 in mesenchymal cells, and 446 in osteogenic cells at a false discovery rate (**FDR**) < 0.01 (**Figs 3A and S14, and S11 Table**). We considered the overlap between interspecific DE genes identified at different stages, initially by performing a pairwise comparison (**S15 Fig**). Although this pairwise approach is straightforward, it is not designed to capture dependence among multiple experimental conditions, and it is not ideal for detecting genes that are consistently DE but have small effect sizes. To address these issues, we also used Cormotif [45] to implement a Bayesian clustering approach capable of capturing the major patterns of correlation between interspecific DE genes identified at different stages of differentiation (**Methods and S10 Table**). Two common temporal expression patterns (or correlation motifs) best fit our pseudobulk data (**Fig 3B**). One motif notes a high degree of interspecific DE genes shared across all stages of differentiation, while a second contains interspecific DE genes unique to pluripotent cells. Using Cormotif, we detected 6,822 interspecific DE genes in pluripotent cells, 4,523 in mesenchymal cells, and 4,020 in osteogenic cells with a posterior probability > 0.65 (**Figs 3B, S16, S17, and S12 Table**). However, due to the high degree of sharing across stages of differentiation, only 2,759 are unique to pluripotent cells, 164 are unique to mesenchymal cells, and none are unique to osteogenic cells (**S16 Fig**). Overall, there is a decrease in interspecific DE genes as cells mature.

After measuring the degree to which interspecific DE genes are shared across the three stages of differentiation, we used Cormotif [45] to identify genes that are stage-specific; that is, genes that are DE between subsequent stages of differentiation. We then asked whether stage-

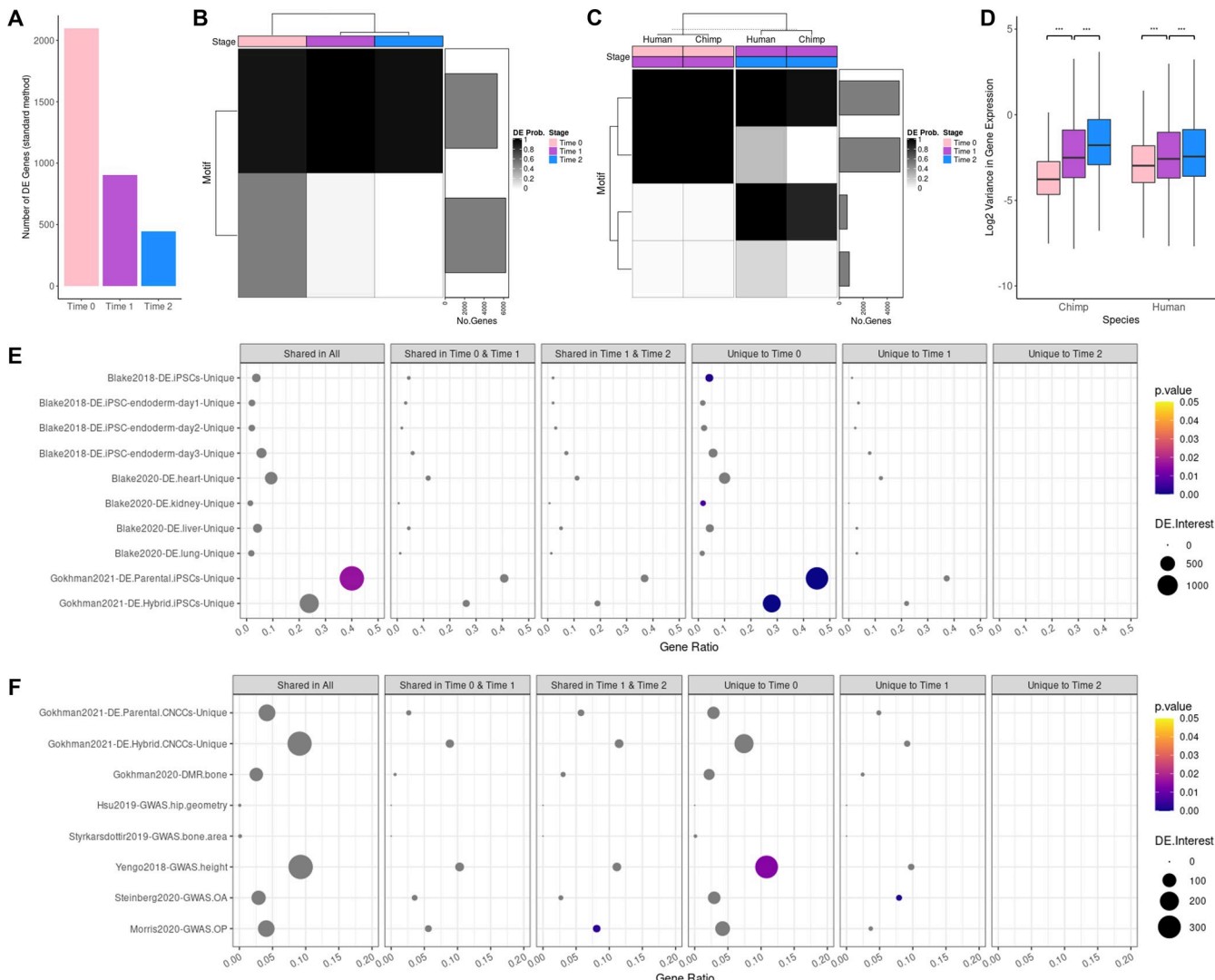

**Fig 3. Interspecific DE across three stages of osteogenic differentiation.** Bar plot showing the number of interspecific DE genes identified for each stage of differentiation using standard methods (A). Correlation motifs based on the probability of differential expression between species for each stage of differentiation (B) and correlation motifs based on the probability of differential expression across stages of differentiations for each species (C) with the number of genes assigned to each motif shown in the bar plot on the right and the posterior probability that a gene is DE shown by the shading of each box. Box plots of the log2 transformed gene expression variance values for cells collected at each stage of differentiation for each species (*** p<0.001) (D). Enrichment of external DE gene sets among Cormotif interspecific DE genes identified for each stage of differentiation for validation (E) and functional interpretation (F) with the p-value (p.value), the number of DE genes overlapping an external gene set (DE.Interest), and the ratio of overlapping to non-overlapping DE genes for a given external gene set (GeneRatio) denoted.

specific genes are generally conserved among humans and chimpanzees. We identified four correlation motifs that best fit our pseudobulk data (**Fig 3C**) and found that the majority (about 89%) of stage-specific DE genes are conserved between species (**S13 Table**). Of the remaining divergent stage-specific DE genes, we detected 20 between pluripotent and mesenchymal cells and 266 between mesenchymal and osteogenic cells, which was somewhat unexpected given that there are substantially fewer stage-specific DE genes at later stages of differentiation than at earlier stages of differentiation. Additionally, this finding was surprising given our previous observation that there are fewer interspecific DE genes in osteogenic cells, which had initially suggested to us that gene expression patterns are more conserved in

osteogenic cells than in early-stage cells. Another possible explanation for the observed decrease in interspecific DE over time is that DE is more difficult to detect in late-stage cells due to increased levels of gene expression variation.

Indeed, examining a total of 11,579 genes, we found a significant increase in gene expression variance between pluripotent cells and mesenchymal cells and between mesenchymal cells and osteogenic cells–a pattern that is maintained across species (**Figs 3D and S18**). However, since there is no *a priori* reason to expect mesenchymal and osteogenic cells to have higher gene expression variance than other cell types, we hypothesized that the high variance we observe is more likely due to increased cell heterogeneity over the course of differentiation. Thus, our subsequent analyses specifically address this property of the data.

### Interspecific differential expression across five substages of osteogenesis

Our single-cell data allowed us to explore how cellular heterogeneity changes throughout differentiation. Although osteogenic differentiations were designed to push cells toward later stages of osteogenesis, we did not expect that all cells would develop into mature osteocytes. Rather, we anticipated that cells would reach variable stages of osteogenesis at the time of collection, either because differentiation started earlier or later in different cells, or because differentiation occurred at different rates in different cells. Indeed, increased gene expression variance in osteogenic cells (**Figs 3D and S18**) hints at such cellular heterogeneity. Thus, we explored several methods to more precisely define each cell's position along the course of osteogenesis (**S1 Text**). We were particularly interested in whether a different method of classifying cells would impact the classification of interspecific DE; namely, we asked: would classifying cells more precisely reveal differences in the speed of osteogenesis between species, and would doing so help us identify more interspecific DE genes?

Five distinct stages of osteogenesis can be distinguished using the expression levels of marker genes. Compared to osteogenic precursors, osteogenic cells are broadly characterized as having increased expression of collagen I and alkaline phosphatase genes. More specific cell types emerge throughout osteogenesis, beginning with preosteoblast progenitors, transitioning to cells embedding themselves in collagen matrix and subsequently mineralizing that matrix, and finally, once encased in bony matrix, maturing into osteocytes [46]. Using these known stages of osteogenesis that have designated candidate genes [46], we classified cells using an ad hoc approach (**Fig 4, Methods, and S1 Text**), which we found produced more biologically meaningful groups of cells than standard clustering methods (**S19 and S20 Figs and S1 Text**). Using our ad hoc annotation scheme, we detected 2,004 preosteoblasts, 3,878 osteoblasts, 3,019 embedding osteoblasts, 4,473 mineralizing osteoblasts, and 2,443 maturing osteocytes (**Figs 4A and S21**)–assignments that are compatible with the results of our standard cell differentiation staining validations (**Figs 1D and S2**). Although these stages of osteogenesis are unified by general osteogenic gene expression patterns, their pseudobulk transcriptomic profiles become subtly but increasingly divergent the longer osteogenesis proceeds (**S22 Fig**). Additionally, because we assigned cells to specific osteogenic stages using marker gene expression (**Fig 4B**), they do retain their characteristic cell-specific expression patterns (**S19C Fig**).

To determine whether the timing of osteogenic differentiation differs between humans and chimpanzees, we compared the distribution of cell counts at each stage of osteogenesis between species. Interestingly, although humans and chimpanzees have similar numbers of preosteoblasts, the distribution of cell counts at other stages of osteogenesis differs between species. We observed an accumulation of intermediate-stage cells (osteoblasts, embedding osteoblasts, and mineralizing osteoblasts) in chimpanzees. In contrast, humans have a reduction of cells at intermediate stages (embedding osteoblasts) and an increase of cells at later

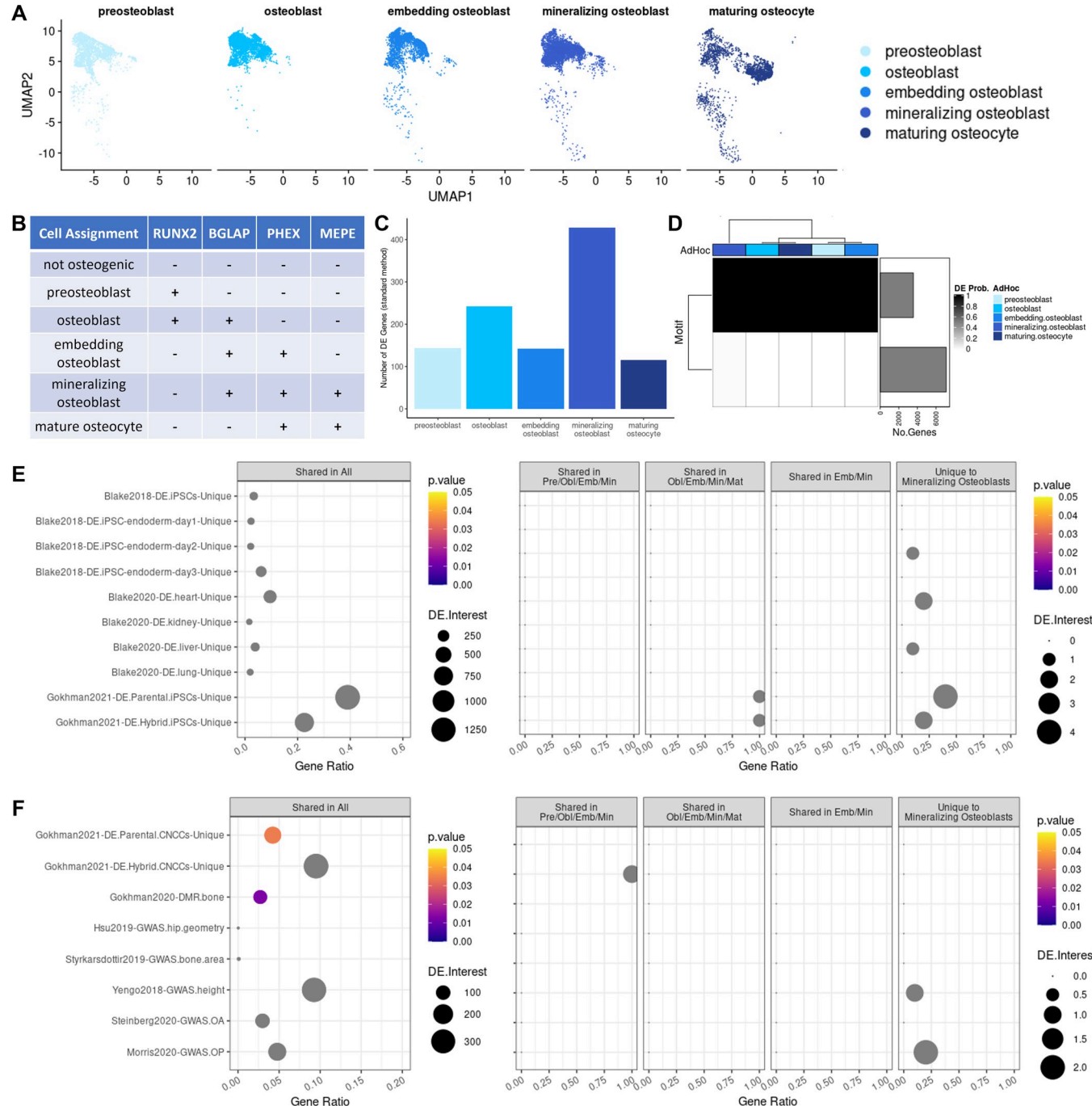

**Fig 4. Interspecific DE across five substages of osteogenesis.** UMAP dimensional reduction plot of scRNA-seq data with cells labeled by the stage of osteogenesis to which they were assigned (A), along with a simplified schematic of the osteogenic ad hoc assignment method. Bar plot showing the number of interspecific DE genes identified for each stage of osteogenesis using standard methods (C). Correlation motifs based on the probability of differential expression between species for each osteogenic ad hoc assignment with the number of genes assigned to each motif shown in the bar plot on the right and the posterior probability that a gene is DE shown by the shading of each box (D). Enrichment of external DE gene sets among Cormotif interspecific DE genes identified for each stage of osteogenesis for validation (E) and functional interpretation (F) with the p-value (p.value), the number of DE genes overlapping an external gene set (DE.Interest), and the ratio of overlapping to non-overlapping DE genes for a given external gene set (GeneRatio) denoted.

stages of osteogenesis (mineralizing osteoblasts and maturing osteocytes) (**S19B Fig**). These cell count distributions reflect the patterns we observed using our more traditional validation methods. For example, on average, differentiating chimpanzee cell cultures had a lower production of calcium deposits than differentiating human cell cultures (**S2 Fig**), consistent with the notion that chimpanzees had a larger proportion of osteogenic precursor cells than humans. These observations suggest that human and chimpanzee cells transition between stages of osteogenesis at different rates.

To assess the extent of interspecific DE between these more nuanced stages of osteogenesis, we performed DE analyses of pseudobulk data (**Methods and S10 Table**). Using standard methods, we identified 144 interspecific DE genes in preosteoblasts, 242 in osteoblasts, 142 in embedding osteoblasts, 429 in mineralizing osteoblasts, and 115 in maturing osteocytes with an FDR < 0.01 (**Figs 4C and S15, and S11 Table**). After accounting for overlaps, this results in a total of 644 interspecific DE genes–many more than were detected in bulk osteogenic cells using our earlier three-stage classification method. As before, we used Cormotif [45] as a second method of identifying interspecific DE genes to better assess sharing across stages of osteogenesis (**Methods**). We found that 2 correlation motifs best fit our pseudobulk data (**Fig 4D**). One motif notes a high degree of interspecific DE gene sharing across stages of osteogenesis, while a second contains genes that show no DE across stages of osteogenesis. Using Cormotif, we detected 3,287 interspecific DE genes in preosteoblasts, 3,287 in osteoblasts, 3,289 in embedding osteoblasts, 3,299 in mineralizing osteoblasts, and 3,287 in maturing osteocytes with a posterior probability > 0.65 (**Figs 4D and S16, and S12 Table**). Almost all of these interspecific DE genes are shared across stages of osteogenesis, with unique interspecific DE genes found only in mineralizing osteoblasts (n = 10) (**S16 Fig**). Overall, this more nuanced approach to classifying cells, which was based on known marker genes, helped us to identify additional interspecific DE genes.

## A continuous model of interspecific differential expression throughout osteogenesis

All of our analyses thus far have implicitly considered cell type to be a discrete phenomenon, despite the fact that cell types are known to be continuous (e.g., distributed from the pluripotent state across intermediate developmental events into mature osteogenic cells). Specifically, our methods have relied on the expression levels of a small number of marker genes to classify cells into types. While this is a standard approach in the field, we felt it was also important to examine continuous variation in our data using information from the whole transcriptome.

We accomplished this with topic modeling, which is an unsupervised classification approach that finds recurring patterns of gene expression within a dataset and then summarizes the expression profile of a cell as a mixture of these identified gene programs (or topics). Because this method allows each cell to have grades of membership in multiple topics simultaneously, we were able to examine both discrete and continuous variation between cells, which is not possible in standard clustering methods (**Figs 5 and S27**). We applied topic modeling (**Methods**) at a range of resolutions, identifying 3, 4, 5, 6, and 7 topics in our data. Additional values of k were considered, and the results are shown in the supplemental data (**S1 Text and S28 and S29 Figs**).

Functional enrichment revealed topics that correspond closely to unsupervised clusters (**Fig 5**), as well as other cell classification schemes that we examined previously. For example, in the 3-topic analysis, pluripotent and mesenchymal cells display distinct gene programs, but this distinction becomes less defined as they transition to osteogenic cells. The dominant topic loadings for each cell classification are enriched in functions (**S14 Table**) that are similar to

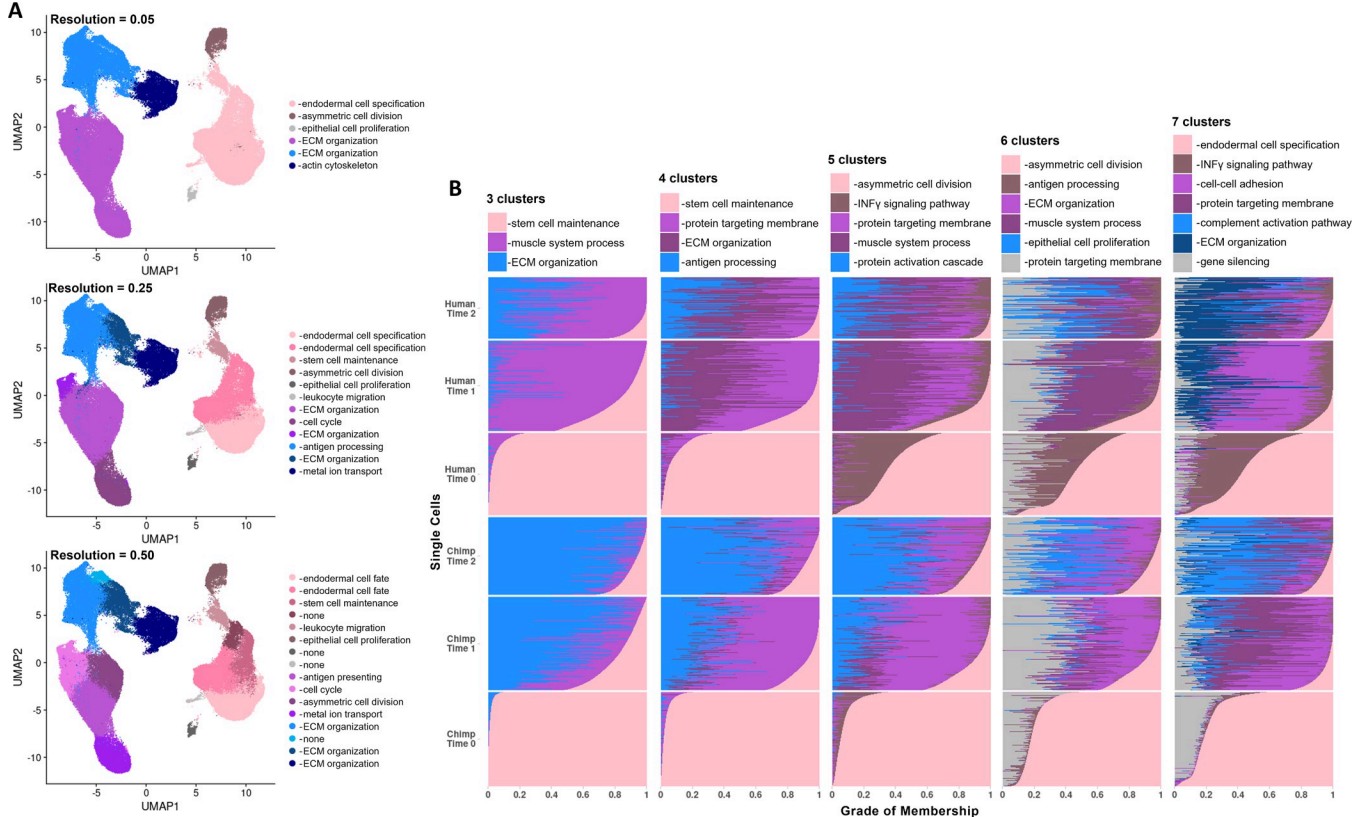

**Fig 5. Examining cell types at different resolutions using discrete and continuous perspectives.** UMAP dimensional reduction plot of scRNA-seq data with cells labeled by the unsupervised cluster (resolutions of 0.05, 0.25, and 0.50) to which they were assigned (A), and structure plots showing the results of topic modeling at k = 3, k = 4, k = 5, k = 6, and k = 7 with each row representing the gene expression profile from one cell, each colored bar representing a topic, and the grade of membership in each topic depicted by the length of the bar along the x-axis (B). In the structure plots, cells are grouped by their species of origin and collection time point. In both sets of plots, the key notes the top GO category enrichment of marker genes for a given cluster or topic.

those identified previously (**Fig 2H, 2I, and 2J**). For instance, in the 3-topic analysis, the dominant topic loading for pluripotent cells (colored pink in **Fig 5B**, k3 in **S27 Fig**) includes genes that are enriched among GO functions related to stem cell maintenance (**S14 Table**), and we find similar functional enrichments for marker genes identified in pluripotent cells (**Fig 2H**).

Interestingly, though, humans and chimpanzees differ in grades of membership as differentiation progresses. Interspecific gene programs, or topics for which different species display different grades of membership, become more pronounced as cells mature. This pattern becomes clearer when we consider differentiation at a more precise scale (or, at increasing values of k). For example, in the 3-topic analysis, while human and chimpanzee pluripotent cells have the same dominant topic and similar grades of membership in this topic, this is not the case for mesenchymal and osteogenic cells. The dominant topics defining mesenchymal and osteogenic cells display interspecific patterns, with humans having greater membership in one topic (colored purple in **Fig 5B**, k1 in **S27 Fig**) and chimpanzees having greater membership in the other topic (colored blue in **Fig 5B**, k2 in **S27 Fig**). This number of gene programs displaying interspecific grades of membership in mesenchymal and osteogenic cells increases at more precise scales, with 3 identified in the 5-topic analysis and 4 identified in the 7-topic analysis (**Figs 5B and S27**).

Although higher values of k also reveal interspecific gene programs at earlier stages of differentiation, the interspecific gene programs at later stages of differentiation are of particular

interest because they are characterized by functions that may provide biological insight into species differences. For example, in the 7-topic analysis, there are two dominant topic loadings for osteogenic cells: one that is chimpanzee-specific (colored light blue in **Fig 5B**, k5 in **S27 Fig**) and one that is human specific (colored dark blue in **Fig 5B**, k3 in **S27 Fig**). While both gene programs are characterized by relevant functions related to extracellular matrix organization (**S14 Table**), only the human-specific gene program has functions related to osteogenesis, including regulation of osteoblast proliferation (P < 0.001), bone trabecula formation (P < 0.001), ossification (P < 0.002), bone maturation (P < 0.003), regulation of biomineral tissue development (P < 0.003), regulation of bone development (P < 0.005), regulation of bone mineralization (P < 0.009), and others (**S14 Table**). These results support our hypothesis that human cells may transition to later stages of osteogenesis more quickly than chimpanzees.

## Concordance of interspecific DE genes with previous studies of differential gene regulation between humans and chimpanzees

Previous studies have identified genes that are differentially regulated between humans and chimpanzees in a number of tissue types [29,36,47,48]. We performed gene set enrichment tests to assess concordance between the interspecific DE genes we identified in this study and previously identified genes. For these analyses, we focused on the interspecific DE genes identified using Cormotif. First, we considered the interspecific DE genes we identified using the three-stage classification approach (**S15 Table** **and Figs 3E, 3F, and S23**). As expected, genes previously identified in iPSCs as DE between species [36,48] are enriched among genes we identified as interspecific DE in pluripotent cells (all P < 0.002) but not among the interspecific DE genes we identified in mesenchymal or osteogenic cells (all P > 0.66). By and large, genes previously identified as DE between species in non-pluripotent, non-mesenchymal, and non-osteogenic cell types and tissues [36,47] are not enriched among genes we identified as DE between species. Still, we found that interspecific DE genes previously identified in kidney tissue [47] are enriched among genes we identified as interspecific DE in pluripotent cells (P < 0.006)–an enrichment that persists regardless of the posterior probability threshold used to classify significant interspecific DE genes (**S17 Fig**). Lastly, while there are no previous studies of gene expression differences between humans and chimpanzees in mesenchymal or osteogenic cells, there has been research on gene expression in related CNCCs [48] and on differentially methylated regions (**DMRs**) in bone tissues [29]. Although we observed overlap of these external gene sets with our interspecific DE genes, we did not identify significant enrichments, even when examining all interspecific DE genes identified in osteogenic cells regardless of sharing (all P > 0.65; **S24 Fig**). However, previously identified interspecific DE genes in CNCCs are slightly enriched among genes we identified as interspecific DE in later stages of differentiation (P < 0.06).

Next, we performed gene set enrichment tests to assess concordance between previously annotated external DE gene sets [29,36,47,48] and the interspecific DE genes we identified across five stages of osteogenesis (**Figs 4E, 4F, and S23 and S15 Table**). As expected, genes previously identified as DE between species in iPSCs [36,48] and in alternative cell types and tissues [36,47] are not enriched among the interspecific DE genes we identified across stages of osteogenesis. Conversely, genes previously found to be differentially regulated between humans and chimpanzees in skeleton-related cell types and tissues do overlap our interspecific DE genes. That is, interspecific DE genes identified in iPSC-derived CNCCs [48] and interspecific DMR-associated genes identified in bone [29] are slightly enriched among interspecific DE genes shared across all stages of osteogenesis (P < 0.03 and P < 0.01, respectively). Again, it appears that there was a benefit to using a higher-resolution osteogenic cell classification

system, as we were able to identify more skeletally relevant functional enrichments when we grouped cells by osteogenic stage rather than bulking them together based on a general collection time point.

## Functional enrichments among interspecific DE genes

To ascertain the biological functions of interspecific DE genes, we examined their overlap with GO categories and relevant GWAS hits, again performing these analyses separately for the two main cell classification strategies used in this study and focusing on the interspecific DE genes identified using Cormotif. First, we considered interspecific DE genes identified using our bulk classification system. We did not find enrichment of interspecific DE genes among skeletally-relevant GO categories (S25 and S26 Figs and S16–S18 Tables). Overlap of skeletal trait and disease-related genes among interspecific DE genes was also limited, and we found no significant overlap between GWAS hits for hip geometry [49] or bone area [50] among interspecific DE genes. Because there are very few genes associated with GWAS hits for hip geometry [49] and bone area [50] that also overlap with genes tested in this study (n = 7 and 8 respectively; S1 Text), we also examined overlapping genes individually. *PPP6R3* and *GAL*, which are associated with hip geometry [49], are interspecific DE genes that are shared across all stages of differentiation. Additionally, of the bone area-associated genes [50] that we tested, *BCKDHB*, *COL11A1*, *CTDSP2* and *SOX9* are interspecific DE genes that are shared between all stages of differentiation; and *BCKDHB*, *COL11A1*, *DYM*, *HHIP*, and *SH3GL3* are interspecific DE genes that are shared between early stages of differentiation (pluripotent and mesenchymal cells). We also examined overlap between interspecific DE genes and broader skeletal phenotypes. We detected significant enrichments of height GWAS hits [51] among interspecific DE genes identified in pluripotent cells (Time 0; P < 0.01), osteoporosis GWAS hits [52] among interspecific DE genes shared across mesenchymal and osteogenic cells (P < 0.003), and osteoarthritis-related loci [53] among interspecific DE genes identified in mesenchymal cells (P < 0.002).

Finally, we ascertained the biological functions of the interspecific DE genes we identified when we grouped cells into five stages of osteogenesis, again examining their overlap with GO categories and relevant GWAS hits (S25 and S26 Figs and S16 and S17 Tables). We observed that interspecific DE genes shared across all stages of osteogenesis are enriched in functional categories related to skeletal trait development and maintenance, including embryonic skeletal system development (P < 0.01), extracellular matrix (P < 0.007), and collagen-containing extracellular matrix (P < 0.009) (S16 Table). We also found that interspecific DE genes unique to mineralizing osteoblasts are enriched in functions regulating cell morphogenesis (P < 0.009) and cell morphogenesis involved in differentiation (P < 0.009) (S16 Table). However, overlap between skeletal trait and disease-related genes among interspecific DE genes was quite limited overall. We found no significant overlap between hip geometry [49], bone area [50], height [51], osteoporosis [52], or osteoarthritis [53] related loci among interspecific DE genes identified in this study. Again, because there are very few overlapping genes associated with GWAS hits for hip geometry [49] and bone area [50] (n = 6 and 6 respectively; S1 Text), we examined these genes individually. Of the hip geometry genes [49] that were tested in our data, *PPP6R3* is an interspecific DE gene that is shared across all stages of osteogenesis. Of the bone area genes [50] that were tested in our data, we identified *COL11A1*, *CTDSP2*, and *SOX9* as interspecific DE genes that are shared between all stages of osteogenesis. When we considered skeletal phenotypes more broadly, we found a slight enrichment of osteoporosis GWAS hits [52] among interspecific DE genes unique to mineralizing osteoblasts (P < 0.07). Altogether, using a higher-resolution classification of osteogenic cells does improve our ability

to detect biologically meaningful interspecific DE genes. These osteogenic interspecific DE genes hint at some skeletal trait associations, but overall, they may only have a moderate impact on skeletal trait differences between species.

## Discussion

This study presents a comparative primate skeletal cell culture model that we established to study gene expression in the skeleton. Using this model, we differentiated a panel of human and chimpanzee iPSCs through an intermediary MSC state and subsequently into osteogenic cells. To our knowledge, this is the largest panel of iPSC-derived osteogenic cell types in any species to date. Other groups have used similar differentiation protocols [31] as well as more directed differentiation protocols [32] in a small number of model organism cell lines, but this study is the first to use a larger, cross-species panel. In comparative primate functional genomics more broadly, there has been research on iPSC-derived CNCCs [30,48], which are precursor cells that contribute to the development of several tissues in the skull, including skeletal tissues. However, we were interested in specifically studying bone cells and opted for a differentiation strategy that more closely resembles the production of bone cells throughout life.

We characterized gene expression patterns at a single-cell resolution in cell types along the differentiation trajectory from iPSCs to osteogenic cells. We specifically chose to collect single-cell data as opposed to bulk data because cell differentiation does not always produce cell types of interest at 100% purity. While some cell types, like cardiomyocytes, have standard purification steps [54] and good flow cytometry markers to estimate cell purity, osteogenic cells do not. Additionally, degrees of cell heterogeneity can vary substantially across different cell lines. Collecting single-cell data allowed us to avoid these issues and instead examine gene expression in specific cell types of interest, while simultaneously providing us with an opportunity to examine gene expression data through the lens of continuous variation.

Overall, we were able to successfully recover high-quality scRNA-seq data from pluripotent cells, mesenchymal cells, and osteogenic cells from both species. Data were reproducible and balanced across species and cell types, indicating that this is a reliable model for studying gene expression patterns. Moreover, using standard methods alongside scRNA-seq analyses, we validated the successful differentiation of mesenchymal cells and osteogenic cells in both species, which further signals that this is an effective model for specifically studying molecular changes in skeletal cell types. While not unexpected, it is interesting that differentiation efficiency varied across individuals within both species. However, teasing apart individual-specific genetic effects that impact differentiation potential will require further exploration in larger sample sets.

Regardless of cell classification, we observed a decrease in interspecific DE as differentiation progresses, which suggests that expression patterns are more conserved in osteogenic cells than in earlier progenitor cells. Nevertheless, a surprising trend in our interspecific DE genes over the course of osteogenesis is that the greatest number of interspecific DE genes are detected in mineralizing osteoblasts, which are the transition cell type between mid-stage embedding osteoblasts and late-stage maturing osteocytes. Interspecific DE genes unique to mineralizing osteoblasts are enriched in functions related to regulating cell morphogenesis in the context of differentiation, so this may be a biological difference between human and chimpanzee skeletal cells that contributes to complex skeletal trait differences. We considered the possibility that some of the regulatory differences we observed may be due to differences in the number of human and chimpanzee cells that were classified as mineralizing osteoblasts; however, randomly subsampling these to equal numbers of cells before performing DE tests did not change the observed patterns. Looking more closely at the number of cells assigned to each

cell type, it is clear that chimpanzees have an accumulation of cells at intermediate stages of osteogenesis as compared to humans, which have increased numbers of late-stage osteogenic cells. This pattern is present in both our scRNA-seq data and standard staining validation results. It is possible that human cells transition through a mineralizing osteoblast state to a maturing osteocyte state more quickly than chimpanzee cells. Such a phenomenon could have broad phenotypic consequences, and the interspecific DE genes that we identified in mineralizing osteoblasts may contribute to these effects. Specifically, based on the functional enrichment of osteogenic interspecific gene programs in our topic modeling, it may be hypothesized that changes related to biomineral tissue development, which enable mineralization, ossification, and trabecula formation, are driving the osteogenesis progression differences that we observed between species. In order to determine whether osteogenesis timing is indeed regulated by such gene regulation, we would need to examine differentiation trajectories at a higher temporal resolution.

We anticipated that identifying and characterizing DE genes at different stages of osteogenesis would be of interest because it is possible for changes in gene expression along the course of osteogenesis to influence differential skeletal trait development. Although we detected more interspecific DE genes when using a higher-resolution cell classification scheme, we only found slight increases in skeletally relevant functional enrichments among these osteogenesis interspecific DE genes as compared to bulk osteogenic interspecific DE genes. That is, interspecific DE genes in iPSC-derived CNCCs [48] and interspecific DMR-associated genes identified in bone tissue [29] are enriched among genes that we identified as interspecific DE across all stages of osteogenesis, but not among bulk osteogenic interspecific DE genes that we identified. These enrichments are functionally relevant in light of previous work, which has found relationships between these external gene sets and divergent skeletal phenotypes between humans and chimpanzees, including differential vocal cord positioning, facial protrusion, and others [29,48]. Overlaps with our datasets, which are more apparent in our high-resolution osteogenesis cell types, suggest that similar skeletal-trait associations exist in our data. Of note, this prior comparative skeletal gene regulation research [29,48] focused on only a handful of genes that may be phenotypically important, including *EVC2*, *NFIX*, *XYLT1*, *ACAN*, *COL2A1*, and *SOX9*. We similarly found that *EVC2, XYLT1*, and *SOX9* are interspecific DE genes shared across stages of osteogenesis, which further supports our intuition that interspecific DE patterns contribute to phenotypic differences. However, *EVC2* and *SOX9* are also interspecific DE genes shared across all stages of differentiation (pluripotent, mesenchymal, and osteogenic cells). This does not dampen the potential impact of DE on resultant phenotypes, but rather, it suggests that this relationship is not isolated to skeletal cells.

We were surprised that regardless of whether we used a broad or precise method of classifying cells, we did not identify enrichments of skeletal disease related genes among our osteogenic interspecific DE genes. Although not unique to osteogenic cells, we did identify an enrichment of osteoporosis GWAS hits [52] among interspecific DE genes shared across later stages of differentiation (mesenchymal and osteogenic). Osteoporosis, which is characterized by a decrease in bone mineral density that can lead to bone fractures, has been observed in nonhuman primates, and manifestations of this disease appear to vary across different primate species [8–10]. This functional enrichment is reasonable because *in vivo* mesenchymal cells are precursors to osteogenic cells. In particular, this enrichment may signal important differences between humans and chimpanzees regarding bone turnover and susceptibility to osteoporosis.

We also found an enrichment of osteoarthritis-related loci [53] among interspecific DE genes identified in mesenchymal cells. Osteoarthritis is characterized by the degradation of cartilage and the underlying bone in joints, and humans and chimpanzees vary in their susceptibilities to this skeletal disorder. In particular, osteoarthritis is prevalent in humans [4], and

while it is similarly prevalent in other nonhuman primates like baboons [3], the prevalence of osteoarthritis in wild [5] and captive [6,7] chimpanzees is very low. It is interesting that this enrichment was found among interspecific DE genes in mesenchymal cells, as these cells are not the primary cells in tissues directly affected during osteoarthritis progression (e.g., cartilage and bone). However, because mesenchymal cells serve as precursor cells for these skeletal tissues, this enrichment may signal important differences between humans and chimpanzees regarding tissue development and maintenance, as well as susceptibility to osteoarthritis. Overall, given these enrichments in mesenchymal cells, along with those identified in cell types transitioning from mesenchymal to osteogenic states, it is worth looking at earlier stages of mesenchymal cell differentiation and osteogenesis more systematically to better identify skeletal trait related DE genes. Additionally, future research incorporating disease related perturbations will allow us to explore this evolutionarily divergent trait in greater depth and detail.

In conclusion, we have established a novel comparative primate skeletal cell culture model that can be used to examine gene expression and various aspects of primate skeletal gene regulation. This includes regulatory mechanisms such as histone modifications and DNA methylation, regulatory responses to environmental treatments, and more. Our findings reveal novel information about the evolutionary changes in gene expression patterns across primate skeletal cell development, and there are many other unexplored research avenues which this system is poised to address.

## Methods

### Ethics statement

iPSC lines used in this study were previously generated [38] from fibroblasts collected from human participants with written informed consent under the University of Chicago IRB protocol 11–0524 and from fibroblasts collected from chimpanzees at the Yerkes Primate Research Center of Emory University under protocol 006–12, in full compliance with IACUC protocols and prior to the September 2015 implementation of Fish and Wildlife Service rule 80 FR 34499.

### Human and chimpanzee iPSC lines

This study included six iPSC lines from humans and six iPSC lines from chimpanzees (**Fig 1B and S1 and S2 Tables**), which is a sufficient number to identify interspecific gene expression differences [35–39]. Technical replicates (independent MSC and osteogenic differentiations) from one human cell line and one chimpanzee cell line were used to examine experimental reproducibility. iPSC lines were derived from fibroblasts using the same experimental design and episomal reprogramming protocol as previously described [38], and pluripotency was previously characterized [34–39]. In addition to species having matched cell type of origin and reprogramming method, biological replicates within each species comprise an equal sampling of both sexes (**Fig 1B and S1 and S2 Tables**).

### Cell differentiation protocols

Feeder-free iPSCs were maintained on Matrigel Growth Factor Reduced Matrix (354230, Corning, Bedford, MA, USA) at a 1:100 dilution and in mTeSR1 Medium (85851, STEMCELL Technologies, Vancouver, Canada) supplemented with 1% Penicillin/Streptomycin (30-002-CI, Corning). Cells were cultured at 37˚C in 5% $CO_2$ and atmospheric $O_2$ and passaged every 3–4 days using a dissociation reagent (0.25mM EDTA, 150mM NaCl in PBS).

iPSC-derived MSCs and osteogenic cells from both species were differentiated using protocols modified from [31] (**Fig 1A and 1C**). After iPSCs were cultured for 15–30 passages as described above, they were seeded at approximately 30% confluency in Matrigel-coated culture dishes and cultured in mTeSR1 (85851, STEMCELL Technologies) until cells fully adhered to the plate (at least 3 hours). Following this, the medium was replaced with MSC medium, which consisted of Dulbecco's Modified Eagle Medium (**DMEM**) (10567–014 or 11330–032, Thermo Fisher Scientific, Waltham, MA, USA) supplemented with 20% fetal bovine serum (**FBS**) (FB5001, Thomas Scientific, Swedesboro, NJ, USA) and 1% Penicillin/Streptomycin (30-002-CI, Corning) (**S2 Table**). Cells were cultured at 37˚C in 5% $CO_2$ and atmospheric $O_2$ with daily MSC medium changes until 80–100% confluent (2–5 days). Cells were subsequently detached from the Matrigel-coated culture dishes using 0.05% Trypsin (25-052-CI, Corning) and passaged to uncoated polystyrene culture dishes. Cells continued to be cultured at 37˚C in 5% $CO_2$ and atmospheric $O_2$ with MSC medium changes every 2–3 days, and cells were subcultured at an approximately 1:3 ratio until at least passage 4, when cells began to display characteristic MSC morphologies (3–9 weeks) (**S3 Table**) [44]. At this point, cells were classified as iPSC-derived MSCs and cryopreserved in cryomedium consisting of 80% FBS, 10% MSC medium, and 10% Dimethyl Sulfoxide (25-950-CQC, Corning).

For osteogenic differentiations, iPSC-derived MSCs were cultured to at least passage 6 and seeded at $4.2 \times 10^4$ cell/$cm^2$ in culture dishes (4-10cm diameter) coated in 5ug/$cm^2$ Type I Collagen (sc-136157, Santa Cruz Biotechnology, Dallas, TX, USA). After culturing cells for 1 day in MSC medium, this medium was replaced with osteogenic medium that contained DMEM (11965–092, Thermo Fisher Scientific) supplemented with 10% FBS (FB5001, Thomas Scientific), 1% Penicillin/Streptomycin (30-002-CI, Corning), 50ug/mL Vitamin C (A4034 or A8960, Sigma-Aldrich, St Louis, MO, USA), 100nM Dexamethasone (D4902 or D1756, Sigma-Aldrich), 10mM β-glycerophosphate (G9422, Sigma-Aldrich), and 1uM Vitamin D (D1530, Sigma-Aldrich). Cells were cultured at 37˚C in 5% $CO_2$ and atmospheric $O_2$, and medium was changed every 2–3 days for 3 weeks.

For all data collections, cells were cultured in discrete batches so that species and cell culture batches were not confounded (**S2 Table**). In each batch, data collections from iPSCs and MSCs were performed after cells were cultured for at least 1 passage following cryostock thawing, and data collections from osteogenic cells were performed immediately after completing the differentiation protocol described above (**S3 and S4 Tables**). iPSC-derived MSCs also underwent chondrogenic and adipogenic differentiations using protocols modified from [31] for cell type validation purposes (**S5–S8 Tables and S1 Text**).

## Single-cell RNA sequencing

scRNA-seq data were obtained from all cell lines at three stages of differentiation–pluripotent cells (Time 0), mesenchymal cells (Time 1), and osteogenic cells (Time 2)–using the 10X Genomics Chromium Single Cell 3' Reagent kit (chemistry v3) (**Fig 1A**) with a pooling strategy that ensured that collection batch and species were not confounded (**S3A Fig and S9 Table**). First, cells were cultured as described above and then dissociated from adherent conditions into single-cell suspensions (**S1 Text**). Cell counts and viability measures were performed for each sample separately (**S3B Fig and S9 Table**). Before single-cell sequencing, samples were pooled such that each pool contained one human sample and one chimpanzee sample at the same stage of differentiation at equal proportions. After cell counts and viability measures were performed for each pooled sample (**S3C Fig and S9 Table**), they were loaded into separate wells of 10X chips with a target collection of 5,000 cells per pooled sample (50% human cells and 50% chimpanzee cells). A total of 21 wells across 21 10X chips were used. Single-cell

cDNA libraries were prepared using the 10X Genomics Chromium Single Cell protocol [40]. Libraries were multiplexed into three batches and sequenced to 100 base pairs, paired-end across 17 lanes on the Illumina HiSeq4000 at the University of Chicago Genomics Core Facility (**S9 Table**). These newly reported data are available on NCBI's Gene Expression Omnibus under accession number GSE181744.

### Single-cell data processing

Sequencing read quality was confirmed using Fast QC, and raw scRNA-seq reads were processed using standard 10X Genomics Cell Ranger 3.1.0 pipelines (Cell Ranger) [40] (**S3D Fig**), with the exception that reads were aligned once to the human genome (hg38) and once to the chimpanzee genome (panTro6) and that a curated set of orthologous exons [55] was used for transcriptome alignment (**S1 Text**). Briefly, 10X cell barcodes and UMIs were extracted from reads, and the remaining reads were mapped to genes using the specified reference genomes.

10X cell barcodes were also bioinformatically reassigned to their species of origin using standard Cell Ranger pipelines (**S3E Fig**). As before, reads were aligned to both human (hg38) and chimpanzee (panTro6) genomes, and the same curated set of orthologous exons [55] was used. Additional modifications to the assignment protocol were also incorporated (**S1 Text**). Briefly, in the standard pipeline, Cell Ranger aligns reads to each of two genomes, retaining only those reads that specifically align to one genome and discarding reads that align to both genomes. It then assigns human and chimpanzee cells based on which genome has more aligned UMI counts and assigns cells as multiplets when the human UMI counts and the chimpanzee UMI counts exceed the $10^{th}$ percentile of each species' distribution. These Cell Ranger parameters were originally optimized for mixtures of human and mouse cells, and we determined that they were not ideal for mixtures of human and chimpanzee cells, which have a much lower genetic divergence (**S1 Text**). Therefore, instead of using these default species assignments, cells were assigned as either human or chimpanzee based on the ratio of human-aligned UMIs to the total number of aligned UMIs within each cell (**S4 and S5 Figs**). Specifically, cells with a ratio greater than or equal to 0.9 were assigned as human, and cells with a ratio less than or equal to 0.1 were assigned as chimpanzee (**S3E Fig**).

Once processed, matrices containing gene counts per 10X barcode were imported into R using the Seurat package (v3.1.2) [42,43], and cells assigned as multiplets were removed. Of note, gene counts for cells assigned as human were determined using read alignments to the human genome, and cells assigned as chimpanzee were determined using read alignments to the chimpanzee genome. Cells were filtered if fewer than 1000 UMIs were detected, if fewer than 700 genes were detected, and if more than 25% of reads mapped to the mitochondria. This resulted in a total of 101,000 cells that were used in subsequent analyses.

### Single-cell data integration

After processing and filtering, scRNA-seq data were integrated using Seurat (**S1 Text**). Briefly, cells derived from individual cell lines were treated as individual datasets (n = 14). Data were log normalized, and datasets were integrated across individuals and collection batches using the reference-based, reciprocal PCA method in Seurat [42,43] (**S1 Text**). Integration was performed across all genes that had non-zero UMI counts (n = 18,482). Following integration, dimensional reduction was performed using a UMAP [41] of all principal components that explain more than 0.1% of data variance (n = 13) (**Fig 2A**), and unwanted variation due to UMI counts and the percent of mitochondria-mapped reads were regressed out from the data.

### Single-cell data annotation

Cell type classifications were defined and assigned to single-cell data using several methods (**S1 Text**). The main methods reported here include the stage of differentiation at which cells were collected, unsupervised clustering, and ad hoc assignments, although further modifications to parameters in each of these classification schemes were also examined (**S1 Text**). The stage of differentiation at which cells were collected was known from the experimental design, but all other annotation methods involved additional analytical steps.

Unsupervised clustering was performed in Seurat. Specifically, the 40 nearest neighbors of each cell were determined using the FindNeighbors function and all principal components that explain more than 0.1% of data variance (n = 13). Using this nearest neighbor information, clusters were then identified using the FindClusters function with resolutions of 0.05, 0.25, and 0.50.

Ad hoc assignments were determined using candidate gene expression patterns (**S1 Text**). A cell was defined as positively expressing a particular gene if the expression level for that gene within the cell was greater than the mean expression level of that gene across all cells. Expression levels were based on scaled integrated data that had confounding variables regressed out as described above. First, general ad hoc assignments of pluripotent, mesenchymal, and osteogenic were determined for cells (**S1 Text**), and then more specific ad hoc assignments for different stages of osteogenesis were determined for osteogenic cells. Different stages of osteogenesis include preosteoblasts, osteoblasts, embedding osteoblasts, mineralizing osteoblasts, and maturing osteocytes, and candidate genes known to vary in expression levels across these stages of osteogenesis were examined [46]. Based on additional analyses (**S1 Text**), the final set of candidate genes for defining osteogenic ad hoc assignments included *RUNX2*, *BGLAP*, *PHEX*, and *MEPE* (**Fig 4B**).

### Topic modeling of single-cell data

In addition to annotating single-cell data into discrete cell classifications, data structure was further examined using topic modeling. In this method, major patterns in gene expression (or topics) within the data are learned, and each cell is modeled as a combination of these topics with different grades of membership in each topic. First, raw scRNA-seq counts were filtered to remove genes with 0 counts across all cells and batch corrected for collection and replicate using the BatchCorrectedCounts function in the R package CountClust [56]. Unsupervised topic modeling was then performed using the R package FastTopics [57]. The fit_poisson_nmf function with default parameters was used to fit a Poisson non-negative matrix factorization model with 3, 4, 5, 6, or 7 ranks to the data. Additional ranks were also considered (**S1 Text**). To convert these Poisson non-negative matrix factorization models into a topic model, the fitted loadings matrices were rescaled to a total sum of 1 across each cell barcode and defined as topic probabilities, and the factors matrices were rescaled to a total sum of 1 across each gene and defined as the word probabilities in the resulting topic model.

### Identifying marker genes

For cell annotations, marker genes were identified for each classification using the FindMarkers function in Seurat. In this method, only genes that were detected in at least 25% of cells within a given classification and that had an average log fold change of at least 0.25 were tested. Wilcoxon Rank Sum tests between specific cell classifications (e.g., Time 0) and all remaining cells were performed to determine marker genes, and only those genes that had elevated expression levels in the specific cell classification as compared to all remaining cells were retained. For topic modeling, the top 100 genes defining each topic were identified using the

ExtractTopFeatures function in the R package CountClust [56] in order to interpret the cellular functions captured by each topic.

## Differential expression using standard methods

For cell annotations, DE between species was calculated using the dream package in R [58]. First, pseudobulk expression values were calculated for each unique grouping of cell line, replicate, and cell classification (e.g., cells from C1, replicate 2, that were collected at pluripotent Time 0). Pseudobulk expression values were defined as the sum of gene counts for a particular group. Mitochondrial genes and ribosomal genes were filtered from data, along with genes that had an average log2 transformed counts per million (**CPM**) less than or equal to zero (**S10 Table**). Pseudobulk expression values were then TMM-normalized. In order to account for the mean-variance relationship in these data [59], weighted gene expression values were estimated using the voomWithDreamWeights function. A linear mixed model (**Eq 1**) in which species was modeled as a fixed effect and individual cell line was modeled as a random effect was then fit to the data using the dream function. Multiple testing was corrected using the Benjamini-Hochberg FDR [60], and genes were designated as significantly DE between species if they had an FDR < 0.01. This cutoff was chosen as it optimized expected patterns of gene set enrichments (**S1 Text** and **S14 Fig**).

$$Y \sim \beta_0 + \beta_{species} * X_{species} + \beta_{cell\ line} * X_{cell\ line} + \varepsilon \qquad \text{Eq1}$$

## Differential expression using cormotif

For cell annotations, the Cormotif package in R [45] was used as a secondary method of identifying DE between species and the primary method for identifying DE across differentiation trajectories. This method is ideal as it helps to overcome issues of incomplete power that affect naïve pairwise DE comparisons and to account for dependency in data coming from different stages of differentiation. Specifically, Cormotif implements a Bayesian clustering approach that identifies common temporal expression patterns (or correlation motifs) that best fit the given data and designated pairwise differential tests. First, pseudobulk expression values were calculated as described above, and mitochondrial genes, ribosomal genes, and genes that had an average log2 transformed CPM less than or equal to zero were removed from the data (**S10 Table**). Of note, Cormotif was initially used to analyze microarray data, so standard analyses simply use log2 transformed CPM values without accounting for the mean-variance relationship in these data. However, this is necessary for RNA sequencing data [59], so before identifying correlation motifs, Cormotif functions were modified to first TMM-normalize raw pseudo bulk expression values and estimate weighted gene expression values using the voom function in the R package limma [61]. Genes with posterior probability > 0.65 were defined as significantly DE. This cutoff was chosen as it optimized patterns of external gene set enrichments (**S1 Text** and **S17 Fig**).

Additionally, when examining the conservation of cell classification DE genes across species, DE genes were classified as conserved if the absolute difference in posterior probability between humans and chimpanzees was less than or equal to 0.3, as previously reported [36].

## Gene expression variance

For cell annotations, the variance in gene expression across cell classifications was examined. This was done separately for each species. First, pseudobulk expression values were calculated as described above, and mitochondrial genes, ribosomal genes, and genes that had an average log2 transformed CPM less than or equal to zero were removed from the data (**S10 Table**). For

each gene, mean expression levels and variance in expression were calculated across cells from a given species and cell classification. Finally, two-sided t-tests were performed separately for the distribution of mean values and the distribution of variance values between cell classifications that are adjacent along a given differentiation trajectory (e.g., Time 0 vs. Time 1, Time 1 vs. Time 2, etc.). Plotted variance values were log2 transformed.

### Concordance and functional enrichment tests

The enrichment of GO categories was assessed among various sets of genes identified in this study using the enrichGO function in the clusterProfiler v3.12.0 package in R [62]. This method uses a hypergeometric test to evaluate the enrichment of GO categories in a set of genes as compared to the total genes examined. Enrichment was classified as significant using a p-value cutoff of 0.01 and a q-value cutoff of 0.05. For cell annotation marker genes, only the top 100 marker genes based on average log fold change values were tested for GO enrichment.

 The enrichment of external gene sets was also assessed in our DE gene sets using a Fisher's exact test. In all enrichment tests, the background gene set consisted of all genes that were tested for DE (**S10 Table**). Several external gene sets were manually compiled from papers, including genes previously identified as DE between humans and chimpanzee [29,36,47,48], skeletal phenotype-related genes [49–51], and genes associated with osteoarthritis and osteoporosis [52,53] (**S15 Table**). Genes previously identified as DE between humans and chimpanzee [29,36,47,48] were primarily used to confirm consistent and biologically meaningful DE. One dataset comprised DE genes in human and chimpanzee iPSCs differentiated towards endoderm [36], another comprised DE genes in human and chimpanzee heart, kidney, liver, and lung tissue [47], and a final comprised DE genes in human and chimpanzee parental and hybrid iPSCs differentiated towards CNCCs [48]. In each of these datasets, only DE genes that are unique to each cell or tissue type examined within a given study were considered. An additional dataset comprising DMRs identified between chimpanzees, anatomically modern humans, and archaic hominins in bone tissues [29] was also considered, since DMRs can impact gene expression patterns. In particular, only those genes with DMRs that differentiate anatomically modern humans and chimpanzees were considered. General skeleton related and skeletal phenotype related genes [49–51] and genes associated with the skeletal diseases osteoarthritis and osteoporosis [52,53] were used to assess whether DE genes in this study may influence specific skeletal traits. Datasets included genes associated with hip geometry GWAS loci [49], bone area GWAS loci [50], height GWAS loci [51], and osteoporosis GWAS loci [52], as well as genes that have cross-omics evidence for involvement in osteoarthritis progression [53].

## Supporting information

**S1 Text. Supplemental methods and results.**
(PDF)

**S1 Table. Details of samples used in the study.** References are provided in a separate tab.
(XLSX)

**S2 Table. Known batch information for samples used in the study.** Additional descriptions for each column heading are provided in **S2 Table** (labels).
(XLSX)

**S3 Table. Cell culture details for the MSC samples used in the study.**
(XLSX)

**S4 Table. Cell culture details for all samples used in the study.**
(XLSX)

**S5 Table. Details and results of the flow cytometry analyses.**
(XLSX)

**S6 Table. Details and results of the Alizarin Red S analyses.**
(XLSX)

**S7 Table. Details and results of the Oil Red O analyses.**
(XLSX)

**S8 Table. Details and results of the qPCR analyses.**
(XLSX)

**S9 Table. Known batch information for 10X single-cell sequencing collections performed in the study.** Additional descriptions for each column heading are provided in **S9 Table (labels)**.
(XLSX)

**S10 Table. The numbers of filtered genes tested in standard and Cormotif DE analyses across cell classifications.**
(XLSX)

**S11 Table. Standard DE genes identified between species for given cell classifications.**
(XLSX)

**S12 Table. Cormotif DE genes identified between species for given cell classifications.**
(XLSX)

**S13 Table. Cormotif DE genes identified between cell types for given cell classifications.**
(XLSX)

**S14 Table. Significant enrichment of GO functional categories in the top 100 genes defining gene programs identified in the topic modeling analyses.**
(XLSX)

**S15 Table. Summary of the external gene sets compiled for functional enrichment tests.** References are provided in a separate tab.
(XLSX)

**S16 Table. Significant enrichment of GO functional categories in Cormotif DE genes identified for given cell classifications and interspecific DE gene sets.**
(XLSX)

**S17 Table. Significant enrichment of GO functional categories in standard DE genes identified for given cell classifications and interspecific DE gene sets.**
(XLSX)

**S18 Table. Significant enrichment of GO functional categories in Cormotif DE genes identified for given cell classifications and cell type DE gene sets.**
(XLSX)

**S1 Fig. Validation of MSC Differentiation.** (A) Phase contrast imaging at 4X of each cell line when pluripotent (Time 0, top row) and mesenchymal (Time 1, bottom row). (B) Box plots showing the proportion of cells expressing CD90, CD73, CD105, or control markers (CD45, CD34, CD14 or CD11b, CD19, and HLA-DR) for pluripotent cells (Time 0, pink) and

mesenchymal cells (Time 1, purple) from all human biological and technical replicates (left set of plots) and all chimpanzee biological and technical replicates (right set of plots). Statistical significance was determined using one-sided Mann Whitney tests. (C) Representative flow cytometry analysis in one human cell line (left) and one chimpanzee cell line (right). Each set of plots displays results for the isotype control (top row), pluripotent cells (Time 0, second row), and mesenchymal cells (Time 1, bottom row). Plots in each row from left to right: (1) forward scatter area versus side scatter area of total cells, (2) forward scatter area versus forward scatter height of Gate 1 cells, (3) Zombie Violet fluorescence versus side scatter area of Gate 2 cells, (4) CD90 FITC fluorescence versus side scatter area of Gate 3 cells, (5) CD73 APC fluorescence versus side scatter area of Gate 3 cells, (6) CD105 PerCPCy5.5 fluorescence versus side scatter area of Gate 3 cells, (7) control marker (CD45, CD34, CD14 or CD11b, CD19, and HLA-DR) PE fluorescence versus side scatter area of Gate 3 cells. (D) Histological validation of MSC differentiation potential in human cell lines (left set of plots) and chimpanzee cell lines (right set of plots). Plots in each set from left to right: (1) images of mesenchymal cells (Time 1) and osteogenic cells (Time 2) after Alizarin Red staining (image zoomed out to display the entire cell culture well), (2) box plot of absorbance values of extracted Alizarin Red stain from all biological and technical replicates from a given species, (3) phase contrast image at 4X magnification of mesenchymal cells (Time 1) and adipogenic cells after Oil Red staining, (4) box plot of absorbance values of extracted Oil Red stain from all biological and technical replicates from a given species, (5) images of mesenchymal cells (Time 1) and chondrogenic cells after Alcian Blue staining (image zoomed out to display the entire cell culture well). (E) Box plots showing the relative quantification levels calculated using the $E^{-\Delta\Delta C_T}$ method of osteogenic (*COL1A1*, *BGLAP*, *RUNX2*), adipogenic (*FABP4*, *PPARG*), and chondrogenic (*COL2A1*, *SOX9*, *COL11A1*) candidate genes in mesenchymal cells (Time 1) and osteogenic cells (Time 2) in all samples. Statistical significance was determined using one-sided t-tests on ΔΔCt values. Box plots: middle line marks the median, box outlines the first and third quartiles, whiskers extend to 1.5 times the interquartile range. Significance: NS. $p > 0.05$, * $p < 0.05$, ** $p < 0.01$, *** $p < 0.001$.
(TIF)

**S2 Fig. Validation of Osteogenic Differentiation.** (A) Imaging of Alizarin Red stain in cell types from each cell line. Images in each row from top to bottom: (1) stained mesenchymal cells (Time 1) zoomed out to display the entire cell culture well, (2) stained osteogenic cells (Time 2) zoomed out to display the entire cell culture well, (3) phase contrast imaging at 4X of stained osteogenic cells (Time 2). (B) Box plots showing the absorbance values of extracted Alizarin Red stain from all biological and technical replicates. Statistical significance was determined using one-sided Mann Whitney tests. Box plots: middle line marks the median, box outlines the first and third quartiles, whiskers extend to 1.5 times the interquartile range. Significance: NS. $p > 0.05$, * $p < 0.05$, ** $p < 0.01$, *** $p < 0.001$.
(TIF)

**S3 Fig. 10X Single-Cell Collections.** (A) Schematic of the dissociation and pooling protocol for 10X single-cell collections. (B) Box plots show cell viabilities for each species and cell type after disassociation. (C) Box plots show cell viabilities for pooled samples of human and chimpanzee cells after mixing. (D) Proportion of mapped scRNA-seq reads. (E) Proportions of species assignments for each 10X collection using Cell Ranger. Box plots: middle line marks the median, box outlines the first and third quartiles, whiskers extend to 1.5 times the interquartile range. Silhouette images were adapted from http://phylopic.org/ and courtesy of T. Michael Keesey and Tony Hisgett (http://creativecommons.org/licenses/by/3.0/).
(TIF)

**S4 Fig. Testing Cell Ranger Species Assignments.** Results of running Cell Ranger on one dataset that contained only human cells [YG-AH-2S-ANT-1] (A,D,G), one dataset that contained only chimpanzee cells [SRR8403265] (B,E,H), and one dataset that contained only human cells [SRR8403264] (C,F,I). Top plots show species assignments for each cell when using the standard Cell Ranger pipeline (A,B,C). Plots in each set from left to right: (1) UMI counts aligned to the human genome vs. UMI counts aligned to the chimpanzee genome with cells colored by assignment, (2) number of cells called as human, chimpanzee, or multiplet. Middle plot shows a histogram of the ratio of human-aligned UMI counts per cells vs. the total number of aligned UMI counts per cells (bars near 0 are likely chimpanzee cells and bars near 1 are likely human cells) (D,E,F). Bottom plots show an alternative species assignment using low cutoffs (~0.9 for humans and ~0.1 for chimps) based on the ratios shown in the histograms (G,H,I). Plots in each set from left to right: (1) UMI counts aligned to the human genome vs. UMI counts aligned to the chimpanzee genome with cells colored by assignment, (2) number of cells called as human, chimpanzee, or multiplet.
(TIF)

**S5 Fig. Applying Modification to Cell Ranger Species Assignments.** Results of running Cell Ranger vs. our modification on all datasets in the study. Top plots show species assignments for each cell when using the standard Cell Ranger pipeline (A,B,C,J,M,P,S). Plots in each set from left to right: (1) UMI counts aligned to the human genome vs. UMI counts aligned to the chimpanzee genome with cells colored by assignment, (2) number of cells called as human, chimpanzee, or multiplet. Middle plot shows a histogram of the ratio of human-aligned UMI counts per cells vs. the total number of aligned UMI counts per cells (bars near 0 are likely chimpanzee cells and bars near 1 are likely human cells) (D,E,F,K,N,Q,T). Bottom plots show an alternative species assignment using low cutoffs (~0.9 for humans and ~0.1 for chimps) based on the ratios shown in the histograms (G,H,I,L,O,R,U). Plots in each set from left to right: (1) UMI counts aligned to the human genome vs. UMI counts aligned to the chimpanzee genome with cells colored by assignment, (2) number of cells called as human, chimpanzee, or multiplet.
(TIF)

**S6 Fig. Data Integration, Reproducibility, and Candidate Gene Expression.** (A) UMAP dimensional reduction plots of scRNA-seq data with cells labeled by the stage of differentiation at which they were collected and separated by species. (B) The correlation of average gene expression patterns between technical replicates separated by species. Plots from left to right: (1) correlation in pluripotent cells (Time 0), (2) correlation in mesenchymal cells (Time 1), (3) correlation in osteogenic cells (Time 2). (C) Dot plots depicting the scaled average expression (dot color intensity) and the proportion of cells expressing each gene (dot size) of candidate genes (x-axis) at each stage of differentiation (y-axis).
(TIF)

**S7 Fig. Cell Counts and Viability Measures.** (A) Bar plot depicting the number of cells collected at pluripotent Time 0 from each cell line, with the average cell viability measure (post-dissociation, pre-sample-pooling) denoted within the corresponding bar. (B) Bar plot depicting the number of cells collected at mesenchymal Time 1 from each cell line, with the average cell viability measure denoted within the corresponding bar. (C) Bar plot depicting the number of cells collected at osteogenic Time 2 from each cell line, with the average cell viability measure denoted within the corresponding bar.
(TIF)

**S8 Fig. Whole Transcriptome Correlations in General Cell Classifications.** (A) Pairwise correlations of pseudobulk counts for each gene between stages of differentiation. (B) Pairwise correlations of pseudobulk counts for each gene between unsupervised clusters at a resolution of 0.05. (C) Pairwise correlations of pseudobulk counts for each gene between general ad hoc assignments.
(TIF)

**S9 Fig. Data QC in General Unsupervised Cluster Cell Annotations.** (A) UMAP dimensional reduction plot of scRNA-seq data with cells labeled by their assigned unsupervised cluster (resolution = 0.05). (B) Bar plot depicting the number of chimpanzee and human cells assigned to each cluster. (C) Violin plots displaying the distribution of UMI counts per cell for each cluster. (D) Violin plots displaying the distribution of gene counts per cell for each cluster. (E) The correlation of average gene expression patterns between technical replicates (human and chimpanzee) collected in each cluster. (F-H) Enrichment of GO functional categories in marker genes for iPSC.c1 (F), MSC.c1 (G), and Osteogenic.c1 (H). The top 5 GO functions identified in biological processes (BP), cell components (CC), and molecular functions (MF) are displayed along with the adjusted p-value (p-adjust), the number of marker genes overlapping a GO function (Count), and the ratio of overlapping to non-overlapping marker genes for a given GO function (Gene Ratio).
(TIF)

**S10 Fig. Data QC in General Ad Hoc Assignment Cell Annotations.** (A) UMAP dimensional reduction plot of scRNA-seq data with cells labeled by their ad hoc assignment. (B) Bar plot depicting the number of chimpanzee and human cells in each ad hoc assignment. (C) Violin plots displaying the distribution of UMI counts per cell for each ad hoc assignment. (D) Violin plots displaying the distribution of gene counts per cell for each ad hoc assignment. (E) The correlation of average gene expression patterns between technical replicates (human and chimpanzee) collected in each ad hoc assignment. (F-H) Enrichment of GO functional categories in marker genes for iPSC.c1 (F), MSC.c1 (G), and Osteogenic.c1 (H). The top 5 GO functions identified in biological processes (BP), cell components (CC), and molecular functions (MF) are displayed along with the adjusted p-value (p-adjust), the number of marker genes overlapping a GO function (Count), and the ratio of overlapping to non-overlapping marker genes for a given GO function (Gene Ratio).
(TIF)

**S11 Fig. Pluripotent and Osteogenic Marker Gene Expression Patterns in General Unsupervised Cluster Cell Annotations.** (A) Violin plots displaying the distribution of pluripotent marker gene expression levels (*POU5F1*, *TDGF1*, *SOX2*, *EPCAM*) in each unsupervised cluster (resolution = 0.05). (B) Dot plots depicting the scaled average expression (dot color intensity) and the proportion of cells expressing each gene (dot size) of pluripotent marker genes (x-axis) in each cluster (y-axis). (C) Violin plots displaying the distribution of osteogenic marker gene expression levels (*COL1A1*, *COL1A2*, *ALPL*) in each cluster. (D) Dot plots depicting the scaled average expression (dot color intensity) and the proportion of cells expressing each gene (dot size) of osteogenic marker genes (x-axis) in each cluster (y-axis).
(TIF)

**S12 Fig. Interspecific DE in General Unsupervised Cluster Cell Annotations.** (A) Bar plot showing the number of interspecific DE genes identified for each unsupervised cluster (resolution = 0.05) using standard methods. (B-C) Correlation motifs based on the probability of differential expression between species for each cluster (B) and correlation motifs based on the probability of differential expression across clusters (iPSC.c1, MSC.c1, Osteogenic.c1) for each

species (C) with the number of genes assigned to each motif shown in the bar plot on the right and the posterior probability that a gene is DE between two clusters in a given species shown by the shading of each box. (D-E) Enrichment of external DE gene sets among Cormotif interspecific DE genes identified for each cluster with the p-value (p.value), the number of DE genes overlapping an external gene set (DE.Interest), and the ratio of overlapping to non-overlapping DE genes for a given external gene set (Gene Ratio) denoted.
(TIF)

**S13 Fig. Interspecific DE in General Ad Hoc Assignment Cell Annotations.** (A) Bar plot showing the number of standard interspecific DE genes identified for each ad hoc assignment. (B-C) Correlation motifs based on the probability of differential expression between species for each ad hoc assignment (B) and correlation motifs based on the probability of differential expression across ad hoc assignments for each species (C) with the number of genes assigned to each motif shown in the bar plot on the right and the posterior probability that a gene is DE between two clusters in a given species shown by the shading of each box. (D-E) Enrichment of external DE gene sets among Cormotif interspecific DE genes identified for each ad hoc assignment with the p-value (p.value), the number of DE genes overlapping an external gene set (DE.Interest), and the ratio of overlapping to non-overlapping DE genes for a given external gene set (Gene Ratio) denoted.
(TIF)

**S14 Fig. Assessing FDR Thresholds for Calling Standard DE Genes.** Enrichment p-values (p.value) of external DE gene sets among interspecific DE genes identified across stages of differentiation using standard methods and a range of different FDR cutoffs (FDR.Cutoff). A p-value of 0.05 is denoted by a horizontal line on each plot, and significant enrichments (p-value < 0.05) are highlighted in red. External DE gene sets were chosen for validation purposes–(A) previously identified interspecific DE genes in iPSCs are expected to only be enriched in interspecific DE genes unique to pluripotent cells (Time 0), (B) previously identified interspecific DE genes in alternative cell types (non-pluripotent, non-mesenchymal, and non-osteogenic) are not expected to be enriched in any interspecific DE genes identified in this study, and (C) previously identified interspecific DE genes in alternative tissue types (non-pluripotent, non-mesenchymal, and non-osteogenic) are not expected to be enriched in any interspecific DE genes identified in this study.
(TIF)

**S15 Fig. Overlap of Standard Interspecific DE Genes.** (A) Details regarding the intersection of standard interspecific DE genes identified across stages of differentiation. (B) Details regarding the intersection of standard interspecific DE genes identified across general unsupervised clusters (resolution = 0.05). (C) Details regarding the intersection of standard interspecific DE genes identified across general ad hoc assignments. (D) Details regarding the intersection of standard interspecific DE genes identified across osteogenic ad hoc assignments. (E) Details regarding the intersection of standard interspecific DE genes identified across osteogenic unsupervised clusters (resolution = 0.50).
(TIF)

**S16 Fig. Interspecific Cormotif DE Genes.** (A-B) Plots describing Cormotif interspecific DE genes identified across stages of differentiation. (C-D) Plots describing Cormotif interspecific DE genes identified across general unsupervised clusters (resolution = 0.05). (E-F) Plots describing Cormotif interspecific DE genes identified across general ad hoc assignments. (G-H) Plots describing Cormotif interspecific DE genes identified across osteogenic ad hoc assignments. (I-J) Plots describing Cormotif interspecific DE genes identified across

osteogenic unsupervised clusters (resolution = 0.50). Plots within each set from left to right: (1) bar plot showing the number of interspecific DE genes identified using Cormotif for given cell classifications (A,C,E,G,I), (2) details regarding the intersection of DE genes for given cell classification (B,D,F,H,J).
(TIF)

**S17 Fig. Assessing Posterior Probability Thresholds for Calling Cormotif DE Genes.** Enrichment p-values (p.value) of external DE gene sets among Cormotif interspecific DE genes identified across stages of differentiation using a range of different posterior probability cutoffs (Post. Prob. Cutoff). A p-value of 0.05 is denoted by a horizontal line on each plot, and significant enrichments (p-value < 0.05) are highlighted in red. External DE gene sets were chosen for validation purposes–(A) previously identified interspecific DE genes in iPSCs are expected to only be enriched in interspecific DE genes unique to pluripotent cells (Time 0), (B) previously identified interspecific DE genes in alternative cell types (non-pluripotent, non-mesenchymal, and non-osteogenic) are not expected to be enriched in any interspecific DE genes identified in this study, and (C) previously identified interspecific DE genes in alternative tissue types (non-pluripotent, non-mesenchymal, and non-osteogenic) are not expected to be enriched in any interspecific DE genes identified in this study.
(TIF)

**S18 Fig. Gene Expression Means and Variances.** (A,C,E) Box plots of the mean gene expression values for given cell classifications in each species. (B,D,F) Box plots of the log2 transformed gene expression variance values for given cell classifications in each species. Statistical significance was determined using two-sided t-tests. (A-B) Mean and variance values for stages of differentiation. (C-D) Mean and variance values for general unsupervised clusters (resolution = 0.05). (E-F) Mean and variance values for general ad hoc assignments. Box plots: middle line marks the median, box outlines the first and third quartiles, whiskers extend to 1.5 times the interquartile range. Significance: NS. p>0.05, * p<0.05, ** p<0.01, *** p <0.001.
(TIF)

**S19 Fig. Comparison of Osteogenic Cell Classification Schemes.** Plots compare (A-C) the osteogenic ad hoc assignment method to (D-F) the osteogenic unsupervised cluster method (resolution = 0.50). (A) UMAP dimensional reduction plots of scRNA-seq data with cells labeled by the stage of osteogenesis to which they were classified via the osteogenic ad hoc assignment method. (B) Bar plot depicting the number of chimpanzee and human cells for each osteogenic ad hoc assignment. (C) Dot plots depicting the scaled average expression (dot color intensity) and the proportion of cells expressing each gene (dot size) of candidate genes (x-axis) for each osteogenic ad hoc assignment (y-axis). (D) UMAP dimensional reduction plots of scRNA-seq data with cells labeled by the osteogenic unsupervised cluster to which they were assigned. (E) Bar plot depicting the number of chimpanzee and human cells for each osteogenic cluster. (F) Dot plots depicting the scaled average expression (dot color intensity) and the proportion of cells expressing each gene (dot size) of candidate genes (x-axis) for each osteogenic cluster (y-axis).
(TIF)

**S20 Fig. Interspecific DE in Osteogenic Unsupervised Cluster Cell Annotations.** (A) UMAP dimensional reduction plot of scRNA-seq data with cells labeled by the osteogenic unsupervised cluster (resolution = 0.50) to which they were assigned. (B) Bar plot showing the number of interspecific DE genes identified for each osteogenic cluster using standard methods. (C) Correlation motifs based on the probability of differential expression between species for each osteogenic cluster with the number of genes assigned to each motif shown in the bar

plot on the right and the posterior probability that a gene is DE shown by the shading of each box. (D-E) Enrichment of external DE gene sets among Cormotif interspecific DE genes identified for each osteogenic cluster with the p-value (p.value), the number of DE genes overlapping an external gene set (DE.Interest), and the ratio of overlapping to non-overlapping DE genes for a given external gene set (Gene Ratio) denoted.
(TIF)

**S21 Fig. Osteogenic Ad Hoc Cell Counts for each Individual and Replicate.** Bar plots depicting the number of osteogenic ad hoc assignment cells for each individual and replicate.
(TIF)

**S22 Fig. Comparison of Osteogenic Ad Hoc Assignment Methods.** (A,C,E) Schematic for each considered osteogenic ad hoc assignment method and (B,D,F) pairwise correlations of pseudobulk counts for each gene between osteogenic ad hoc assignments. Correlation values highlighted in red indicate a deviation from the expected pattern. (A-B) Plots of the osteogenic ad hoc assignment method using 8 candidate genes. (C-D) Plots of the osteogenic ad hoc assignment method using 6 candidate genes. (E-F) Plots of the osteogenic ad hoc assignment method using 4 candidate genes.
(TIF)

**S23 Fig. Gene Set Enrichments in Standard Interspecific DE Genes.** Enrichment of external DE gene sets among standard interspecific DE genes (FDR<0.01) identified for given cell classifications with the p-value (p.value), the number of DE genes overlapping an external gene set (DE.Interest), and the ratio of overlapping to non-overlapping DE genes for a given external gene set (Gene Ratio) denoted. (A) Enrichments across stages of differentiation. (B) Enrichments across general unsupervised clusters (resolution = 0.05). (C) Enrichments across general ad hoc assignments. (D) Enrichments across osteogenic ad hoc assignments. (E) Enrichments across osteogenic unsupervised clusters (resolution = 0.50).
(TIF)

**S24 Fig. Gene Set Enrichments in All Cormotif Interspecific DE Genes.** (A-B) Enrichment of external DE gene sets among Cormotif interspecific DE genes identified for each stage of differentiation for validation (A) and functional interpretation (B) with the p-value (p.value), the number of DE genes overlapping an external gene set (DE.Interest), and the ratio of overlapping to non-overlapping DE genes for a given external gene set (Gene Ratio) denoted.
(TIF)

**S25 Fig. GO Enrichments in Cormotif Interspecific DE Genes.** Enrichment of GO functional categories among Cormotif interspecific DE genes identified for a given cell classification. The top 5 GO functions identified in biological processes (BP), cell components (CC), and molecular functions (MF) are displayed along with the adjusted p-value (p-adjust), the number of marker genes overlapping a GO function (Count), and the ratio of overlapping to non-overlapping marker genes for a given GO function (Gene Ratio). (A) Enrichments across stages of differentiation. (B) Enrichments across general unsupervised clusters (resolution = 0.05). (C) Enrichments across general ad hoc assignments. (D) Enrichments across osteogenic ad hoc assignments. (E) Enrichments across osteogenic unsupervised clusters (resolution = 0.50).
(TIF)

**S26 Fig. GO Enrichments in Standard Interspecific DE Genes.** Enrichment of GO functional categories among standard interspecific DE genes (FDR<0.01) identified for a given cell classification. The top 5 GO functions identified in biological processes (BP), cell components (CC),

and molecular functions (MF) are displayed along with the adjusted p-value (p-adjust), the number of marker genes overlapping a GO function (Count), and the ratio of overlapping to non-overlapping marker genes for a given GO function (Gene Ratio). (A) Enrichments across stages of differentiation. (B) Enrichments across general unsupervised clusters (resolution = 0.05). (C) Enrichments across general ad hoc assignments. (D) Enrichments across osteogenic ad hoc assignments. (E) Enrichments across osteogenic unsupervised clusters (resolution = 0.50).
(TIF)

**S27 Fig. Topic Grades of Membership.** Boxplots showing the grade of membership results of topic modeling at k = 3, k = 4, k = 5, k = 6, and k = 7. Each topic is plotted separately with the grade of membership of cells in a topic noted along the y-axis. Cells are grouped by their species of origin and collection time point. The color denoted by each k references the color of that topic plotted in **Fig 5**.
(TIF)

**S28 Fig. Topic Models for Additional Values of k.** Structure plots showing the results of topic modeling at k = 8, k = 9, k = 10, k = 15, k = 20, and k = 30 with each row representing the gene expression profile from one cell, each colored bar representing a topic, and the grade of membership in each topic depicted by the length of the bar along the x-axis. Cells are grouped by their species of origin and collection time point, and the key notes the top GO category enrichment of marker genes for a given topic.
(TIF)

**S29 Fig. Topic Grades of Membership for Additional Values of k.** Boxplots showing the grade of membership results of topic modeling at k = 8, k = 9, k = 10, k = 15, k = 20, and k = 30. Each topic is plotted separately with the grade of membership of cells in a topic noted along the y-axis. Cells are grouped by their species of origin and collection time point. The color denoted by each k references the color of that topic plotted in **S28 Fig**.
(TIF)

## Acknowledgments

We thank members of the Gilad lab for helpful discussions, particularly Kenneth Barr, Michelle Ward, and Anthony Hung. We also thank Dr. Heike E. Daldrup-Link and Olga Lenkov at Stanford University for training during the initial optimization of our differentiation protocols, as well as Natalia Gonzales for help editing the paper. The computational resources were provided by the University of Chicago Research Computing Center.

## Author Contributions

**Conceptualization:** Genevieve Housman, Yoav Gilad.

**Data curation:** Genevieve Housman.

**Formal analysis:** Genevieve Housman.

**Funding acquisition:** Genevieve Housman, Yoav Gilad.

**Investigation:** Genevieve Housman, Emilie Briscoe.

**Methodology:** Genevieve Housman.

**Project administration:** Genevieve Housman.

**Resources:** Genevieve Housman, Yoav Gilad.

**Software:** Genevieve Housman.

**Supervision:** Yoav Gilad.

**Validation:** Genevieve Housman.

**Visualization:** Genevieve Housman.

**Writing – original draft:** Genevieve Housman.

**Writing – review & editing:** Genevieve Housman, Emilie Briscoe, Yoav Gilad.

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
