## [Decision Letter · Decision Letter 0]

5 Nov 2021

Dear Dr Housman,

Thank you very much for submitting your Research Article entitled 'Evolutionary insights into primate skeletal gene regulation using a comparative cell culture model' to PLOS Genetics.

The manuscript was fully evaluated at the editorial level and by independent peer reviewers. The reviewers appreciated the attention to an important topic but identified some concerns that we ask you address in a revised manuscript.

We therefore ask you to modify the manuscript according to the review recommendations. Your revisions should address the specific points made by each reviewer.

[LINK]

Yours sincerely,

Lisa Stubbs

Associate Editor

PLOS Genetics

Bret Payseur

Section Editor: Evolution

PLOS Genetics

Reviewer's Responses to Questions

**Comments to the Authors:**

Reviewer #1: The manuscript by Housman et al sets out to demonstrate the value of an interspecific iPSC model of osteogenic cells for comparative functional studies of bone and associated tissues between humans and chimpanzees. The authors analyse data from over 100,000 individual cells spanning three points, and do this thoroughly and carefully, controlling for interspecific differences and technical confounders as best as possible, and taking multiple approaches to cell clustering and to differential expression testing. In many ways, I found this to be a very technical manuscript, shy on major novel biological insights (through no fault of the authors) but very thorough in describing the features of the model and all analyses undertaken. The biggest observation is that the endpoint reached by the two species appears to differ somewhat; however, not enough is done to explore the causes of this observation, even without generating additional data. However, while the manuscript itself is clearly written and easy to follow, some of the supplementary text so dense that I struggled with it at times, which I note below.

This manuscript clearly represents a phenomenal amount of work and data both. Overall, I found it thorough and well-presented, and have limited specific comments to make, as follows:

Major:

1. Line 223: The authors observe a marked increase in expression heterogeneity across cells as their time course progresses, and a reduction in the number of DE genes, which they attribute to this fact.They also observe that the human and the chimpanzee osteogenic cells appear to reach different endpoints, with more latter stage cells in humans than chimps, which stall at intermediate stages. There's a nod at the significance of this finding in the discussions, but I found it unsatisfactory. Are there are clues in the data as to *why* the chimp cells are stalling?

In addition, in figure S3.7 there's consistently higher variation in chimps than in humans, and the correlation of the osteogenic replicate in chimps is lower than in any other comparisons; there's also a lot more dropouts in the human osteogenic cells than the chimps (figure 2D). Are these biological or technical findings? Are they fully artefactual? If so, do the authors have any idea as to why?

2. Topic modelling: The handling of these results, which suggest again that at latter time points the two species are becoming more committed to different developmental trajectories, is somewhat underwhelming. At higher values of k, divergence between the two species increases, especially at latter developmental timepoints. Does this match the greater divergence also observed in the osteogenic clustering results? Alternatively, is there a way to ascertain an 'optimal' (I do not want to say 'true') number of clusters? At the very least, why did the authors not consider values of k > 7 (note that I am not saying they should, but this was the one time in the manuscript that a threshold was not justified).

3. As said above, I greatly appreciate the exhaustively documented analyses presented in this paper. However, I wonder if there are sections that would be better off streamlined, to make it easier to appreciate the depth of work performed. The description of DE testing across different partitions of the data in the supplementary methods (pgs 11-14), for instance, is very repetitive (all sections beginning with "As expected, previously identified interspecific DE genes in iPSCs [13,14] are enriched among genes"), which makes it hard to keep track of what was being discussed in each section. The authors might want to consider whether there is real value in discussing results from methods they set aside as being inferior to their final approach, or if there's a need to duplicate text between the main text and the supplement. Notably, much of page 11 is redundant with the results presented in the main text.

I realize this is partly a stylistic preference, and thus leave the final decision to the authors. I do think it would improve readability.

4. Clustering approaches (supplementary results): Both the iPSC and osteogenic stages are split into multiple clusters by the unsupervised approach, but little attention is paid to the iPSC ones. I accept they are probably not very exciting. I would still like to see why the authors chose to focus on iPSC.c1 to the exclusion of c2 and c3.

Minor comments:

Line 212: is there anything interesting in the 266 stage-specific osteogenic DE genes? Line 353 states there's no GO enrichment, but that doesn't preclude more granular findings.

Figures: Better labelling of axes would be very appreciated (eg indicate in the y axis whether the comparison is Cormotif or dream-based)

Figures: Resolution of most clusterProfiler figures is poor when zoomed in, both in supplement and main text, but zooming in is necessary for readability. This could well be an artefact of file upload through the journal submission system, however. This is also true, more generally, of most labels in main text figures 2-5, and in many of the supplementary figures.

Figure 2E-G, S2.6E, S2.7E: are these showing results across both species at once or only one?

Supplementary text, page 4: I found the failure of Cell Ranger to handle the cross-species results, and the work done to control for this, notable, and wonder if it might not be better off in the main text?

Supplementary text, page 6, unsupervised clustering methods: "a resolution of 0.75 identifies 8 osteogenic clusters. Both of these annotations were considered in downstream analyses." Were these results actually included? If not, authors should remove that part of the sentence, or clarify why they are not shown.

Supplementary text, page 8: Are the number of unknown osteogenic cells for the strict rows correct? Those rows do not add up to the 17,575 cells of the flexible rows.

Supplementary text, page 10: "Unsupervised clustering with a resolution of 0.05 identified 5 clusters – 3 pluripotent clusters (iPSC.c1, iPSC.c2, iPSC.c3), 1 mesenchymal cluster (MSC.c1), and 2 osteogenic clusters (Osteogenic.c1, Osteogenic.c2) (Figure S2.6A)." That is 6 clusters, unless I'm missing something?

Figure S3.4B: Why are iPSC and osteogenic clusters other than c1 excluded?

Figure S3.5: Why doesn't panel D match the clusters in panel C? Why are iPSC and osteogenic clusters other than c1 excluded?

Figure S4.1: legend does not match the number of panels in the figure.

Typos etc:

Line 197: 'a pairwise comparisons' Should be 'comparison'

Line 267: 'chimpanzee cells produced less calcium deposits' Should that not be 'fewer' deposits?

Line 333: 'sightly' should be 'slightly'

Line 435: 'osteoblasts; However,' should be 'however'

Line 568: 'samples were pools' should be 'pooled'

Reviewer #2: Overview: This is a study reporting on single cell transcriptomics data from three stages of differentiation of bone cells – iPSC, mesenchymal cells, osteogenic cells – from human and chimpanzee. They identify genes that are differentially expressed (DE) between stages and/or between species and glean high-level insights from these DE gene sets. Mostly, the insights they arrive at are consistent with existing knowledge of the process, and shows large degree of evolutionary conservation of the differentiation programs, with some divergence as well. The analyses rely on standard approaches for the most part, and do not reveal particular insights into evolutionary divergence of skeletal traits. The advantage of the single cell data is not particularly clear, as similar insights are arrived at using single-cell and using “pseudo-bulk” data (where the single data have been collapsed to mimic bulk data). All in all, the paper presents a useful data resource for the community.

Summary of paper: The authors detected inter-specific differentially expressed (DE) genes for each of three stages of differentiation (. They found many inter-specific DE genes to be common across stages, and the number of stage-exclusive inter-specific DE genes to be far fewer for later stages. They also focused on inter-stage DE genes (found using a published tool called cormotif) and found that among those inter-stage DE genes that are not conserved between species, there is a greater representation of later stages. The authors observed an increased variance in gene expression at later stages and suspected that this is due to increased heterogeneity of data. To follow up on this speculation, they examined single cell data that provided a more nuanced view of “stages” of differentiation than that afforded by the three sampling stages. The authors found more DE genes by using the single-cell data. They also reported on various systems-level characterizations of various DE gene sets, using standard approaches such as GO enrichment and GWAS gene enrichment, as well as more recently proposed, single cell data-specific approaches such as topic modeling. The authors also examine the identified DE gene sets in relation to previously published reports of interspecific DE genes (between human and chimpanzee) and find enrichments in some cases (where the previous studies examined similar cell types). By and large, these comparisons show that the gene sets reported here are relatively unique, as expected from the fact that the cross-species comparison of mesenchymal and osteogenic cells in this work is novel. Various Gene Ontology (GO) enrichments along expected lines are also reported for the DE genes, while no particular enrichments were observed for relevant disease-related genes.

I only have a few minor comments that may help the authors and leave it to the authors to address them:

A major advantage of the examination of single-cell transcriptomes appears to be the “more nuanced approach to classifying cells”. However, the only new insight this seems to have yielded is a greater number of DE genes. Such a claim is hard to assess rigorously since the number of tests that reject the null hypothesis does not really offer an apple-to-apple comparison when performed on totally different data.

The topic-modeling based analysis (using a previously published method) needs to be clarified in the presentation. The current presentation offers a rather abstruse view of the process. For instance, the reader is asked to intuit about the “dominant topic loading for each cell classification” and then ponder the functional “enrichment” of such topic loadings, leaving them uncertain about how strong of a claim can result from such observations. Ultimately, it is not clear what exactly was learnt from the topic modeling, beyond some “enriched” annotations whose precise interpretation is difficult because it is not clear how strong the underlying evidence is.

It seems an overarching conclusion from the topic modeling analysis is that “interspecific gene programs become more pronounced as the cells mature”. What exactly are “interspecific gene programs”? – those conserved between species or those divergent between species?

The relationship between cell stage assignments based on time of collection and those based on marker genes is unclear. If the UMAP plot of Figure 2A is reproduced with colors indicating the latter (more fine-grained) assignment scheme, this will become clear.

**Have all data underlying the figures and results presented in the manuscript been provided?**

Reviewer #1: Yes

Reviewer #2: Yes

PLOS authors have the option to publish the peer review history of their article (what does this mean?). If published, this will include your full peer review and any attached files.

Reviewer #1: No

Reviewer #2: No

---

## [Editor Report · Decision Letter 1]

2 Feb 2022

Dear Dr Housman,

We are pleased to inform you that your manuscript entitled "Evolutionary insights into primate skeletal gene regulation using a comparative cell culture model" has been editorially accepted for publication in PLOS Genetics. Congratulations!

Yours sincerely,

Lisa J. Stubbs, Ph.D.

Associate Editor

PLOS Genetics

Bret Payseur

Section Editor: Evolution

PLOS Genetics

Comments from the reviewers (if applicable):

**Data Deposition**

http://datadryad.org/submit?journalID=pgenetics&manu=PGENETICS-D-21-01328R1

**Press Queries**

---

## [Editor Report · Acceptance letter]

27 Feb 2022

PGENETICS-D-21-01328R1 

Evolutionary insights into primate skeletal gene regulation using a comparative cell culture model 

Dear Dr Housman, 

We are pleased to inform you that your manuscript entitled "Evolutionary insights into primate skeletal gene regulation using a comparative cell culture model" has been formally accepted for publication in PLOS Genetics! Your manuscript is now with our production department and you will be notified of the publication date in due course.

With kind regards,

Orsolya Voros

PLOS Genetics

On behalf of:
